# Novel FFPE proteomics method suggests prolactin induced protein as hormone induced cytoskeleton remodeling spatial biomarker
Jakub Faktor [1] ✉, Sachin Kote [1] ✉, Michal Bienkowski[2], Ted R. Hupp [1] &
Natalia Marek-Trzonkowska[1]

Robotically assisted proteomics provides insights into the regulation of multiple proteins achieving excellent spatial resolution. However, developing an effective method for spatially resolved quantitative proteomics of formalin fixed paraffin embedded tissue (FFPE) in an accessible and economical manner remains challenging. We introduce non-robotic In-insert FFPE proteomics approach, combining glass insert FFPE tissue processing with spatial quantitative data-independent mass spectrometry (DIA). In-insert approach identifies 450 proteins from a 5 µm thick breast FFPE tissue voxel with 50 µm lateral dimensions covering several tens of cells. Furthermore, In-insert approach associated a keratin series and moesin (MOES) with prolactin-induced protein (PIP) indicating their prolactin and/or estrogen regulation. Our data suggest that PIP is a spatial biomarker for hormonally triggered cytoskeletal remodeling, potentially useful for screening hormonally affected hotspots in breast tissue. In-insert proteomics represents an alternative FFPE processing method, requiring minimal laboratory equipment and skills to generate spatial proteotype repositories from FFPE tissue.

Human clinical samples represent an indispensable source of information on novel biomarkers and molecular pathways underlying diseases. FFPE tissue slides are the most commonly available biological samples in the clinic. Nevertheless, their proteomic analysis is complicated, due to paraffin embedding and protein crosslinking introduced during formalin fixation. Classical bottom-up proteomic approaches fail to effectively retrieve proteins from FFPE tissue. Therefore, several FFPE tissue processing protocols have been developed to remove paraffin and reverse the protein crosslinking.

FFPE tissue proteomic processing protocols often rely on strong detergents applied in the presence of heat to de-crosslink and solubilize the proteins[1–6]. Alternatively, strong detergents or chaotropic reagents are applied after the heat induced de-crosslinking[7]. However, the detergents and chaotropes must be removed from the sample prior digestion or liquid chromatography mass spectrometry (LC-MS) analysis which often leads to excessive sample losses. This problem has partly been overcome by implementing mass spectrometry compatible detergents such as Rapigest, PPS silent surfactant, ProteaseMax or by using direct trypsinization[8–12]. Implementing Tandem Mass Tag (TMT) labeling of FFPE subsections, followed by pooling and adding a booster channel substantially improved sensitivity

of protein detection[13,14]. Alternatively, robotic platforms such as Deep Visual Proteomics or NanoPOTS could be nowadays deployed to process the FFPE tissue with almost a single cell resolution[15,16]. However, current FFPE tissue processing protocols are either too complicated and cost ineffective or do not have sufficient sensitivity to resolve FFPE tissue subsections spatially. In consequence, FFPE tissue proteomic analyses often render a blurred proteomic picture representing an average of protein intensities from functionally distinct regions of FFPE tissue.

We present a widely accessible and economical method which represents an alternative to current FFPE processing methods. The method requires minimal laboratory equipment and skills. It is intended for proteomic processing of FFPE tissue mounted on a glass slide in combination with either macrodissection or laser capture microdissection (LCM). Simultaneously, it allows post-acquisition spatial DIA data mining for any target listed in a spectral library. To demonstrate the potential of the methodology we explored the spatially resolved biochemistry of keratins, prolactin, estrogen and PIP in breast tissue highlighting the potential applications of spatial quantitative repositories. This strategy should facilitate the conversion of FFPE tissue slides into publicly available spatial

[1]International Centre for Cancer Vaccine Science, University of Gdansk, Kladki 24, 80-822 Gdansk, Poland. [2]Medical University of Gdansk, University of Gdansk, Mariana Smoluchowskiego 17, 80-214 Gdansk, Poland. ✉e-mail: jakub.faktor@ug.edu.pl; sachin.kote@ug.edu.pl

quantitative proteomic repositories especially in low-budget and less equipped laboratories.

## Results

### Introduction to the In-insert proteomics

An overview of the new In-insert proteomic protocol applied to voxelated FFPE tissue is shown in Fig. 1 and Supplementary Fig. 1a–e, which shows a macrodissection and LCM microdissection of a breast FFPE tissue slide (BFPT) into voxels. The In-insert protocol was developed by unique combination of a glass insert sample preparation in a wet chamber preventing evaporation of microliter volumes during long-term high temperature exposure (95 °C) with steps of modified in-solution digestion, classical FFPE protein digestion, autoPOTS, surfactant-assisted one-pot (SOP-MS) proteomic protocols and label free DIA/ sequential window acquisition of all theoretical mass spectra (SWATH) mass spectrometry[5–7,17,18]. Figure 1 demonstrates the versatility of the glass insert, which allows repeated snap-freezing in liquid nitrogen, bath sonication, centrifugation and de-crosslinking at nearly 100 °C followed by protease digestion at microliter scale. Therefore, In-insert sample preparation of each FFPE voxel could be done within several microliters of a reaction mixture processed in a single glass insert reactor. n-dodecyl-β-D-maltopyranoside (DDM) detergent was used to extract and solubilize proteins while keeping the sample compatible with downstream tryptic digestion and further data-dependent (DDA)/DIA mass spectrometry. In addition, omitting excessive salts during the sample preparation allowed the sample to be injected directly to the trap column without damaging the LC-MS system. Acquired total ion chromatograms (TICs) suggest that peptide peaks are evenly separated, and their ionization is not suppressed (Supplementary Fig. 2a, b). Detailed description of the optimized In-insert protocol is in the Fig. 1, Materials and Methods section and in Supplementary Fig. 1.

### Defining the required amount of FFPE tissue and benchmarking variants of the In-insert method to the intersecting FFPE tissue processing method

Approximately 1100 protein groups (FDR < 1%) per slide from healthy FFPE breast tissue (N = 11) were processed using the method by Weke et al.[19] to assess protein extractability and estimate protein yield. Interestingly, in comparison to the In-insert method a comparable number of protein identifications from each macrodissected voxel was achieved, with an input size at least 20 times smaller than that by Weke et al.[19]. However, the processing effectivity of In-insert method could be better demonstrated in a combination with LCM which allowed processing hematoxylin stained healthy breast voxels of considerably smaller areas (50 × 50 μm, 100 × 100 μm and 200 × 200 μm) and 5 μm thickness. From the Supplementary Fig. 2b–d and Table 1 it is evident that the In-insert method extracted mass spectrometry compatible proteomic samples even from the smallest 50 × 50 μm size, 5-μm thick hematoxylin stained voxel, resulting in almost 450 protein groups/1200 peptides identified (iPROB >0.99) in one of three biological replicates. Table 1, compares In-insert processing methods for voxels retrieved by macrodissection and LCM, with or without hematoxylin FFPE tissue staining, alongside the method by Weke et al. applied to process entire FFPE tissue slides[19].

A literature search to compare the In-insert method to intersecting FFPE processing protocols demonstrates the In-insert's method effectiveness in minimizing material requirements[1–7,9,11]. Moreover, In-insert combination with LCM is comparable to the robotically assisted autoPOTS. More details on benchmarking the In-insert protocol to other intersecting FFPE processing protocols are communicated in discussion.

Further, we conducted a protein intensities comparison of In-insert proteomics data with a publicly available proteomic dataset utilizing TMT labeling of entire FFPE breast carcinoma slides[20], as well as with healthy BFPT prepared using the protocol by Weke et al.[19]. We plotted the total protein spectral count from these three datasets against the rank of protein. Our focus primarily centered on keratins, as they could be contaminants, and we later used them to explore the role of PIP protein. Supplementary Fig. 3 displays the summed spectral counts of selected keratins across three datasets. Supplementary Fig. 3a illustrates the keratin spectral count in a BFPT voxels processed by In-insert method, while Supplementary Fig. 3b shows the tissue of the same origin processed by the protocol published by Weke et al.[19]. Furthermore, Supplementary Fig. 3c shows total keratin spectral count in the publicly available dataset of FFPE breast cancer tissue labeled by TMT approach[20]. The Supplementary Fig. 3 demonstrates that

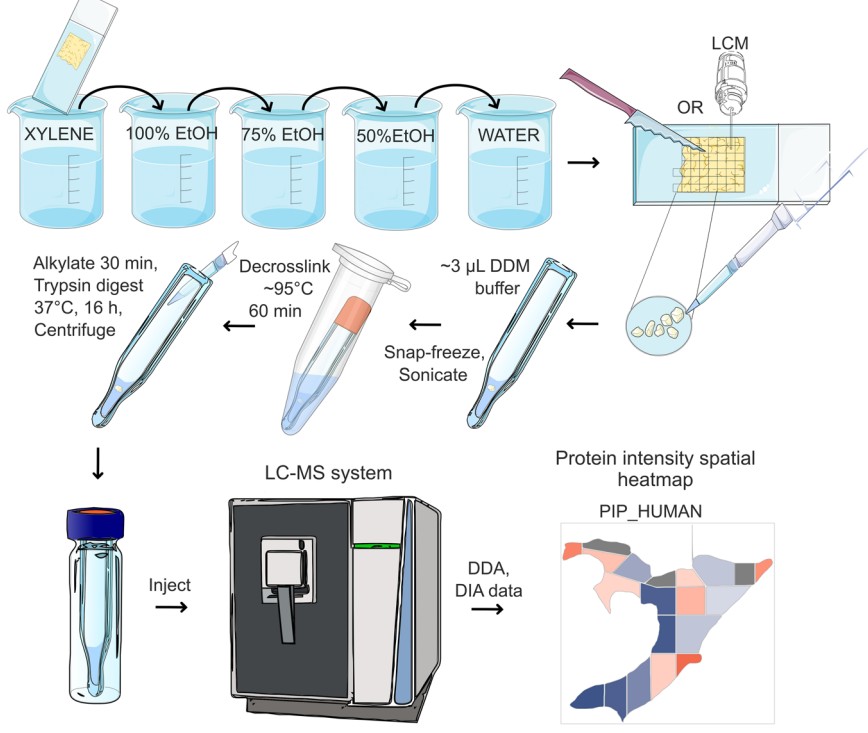

**Fig. 1 | In-insert spatial proteomics workflow.** In-insert proteomics involves processing FFPE sample voxels in a glass insert by transferring the steps of classical protocols to the microliter scale in a wet chamber and omitting mass spectrometry incompatible substances. Low sample loss during In-insert FFPE processing allows spatial proteomic research of biochemical signaling pathways and biomarkers in a glass insert, while keeping low cost of analysis. Implementing DIA/SWATH mass spectrometry enables post-acquisition mining that provides detailed insight into spatial quantitative proteomic signatures within a FFPE slide, relying entirely on publicly available freeware. Freely available Bioicons repository was used to create the modified shapes in this figure (https://bioicons.com/).

**Table 1 | Comparison of experimental results retrieved by In-insert formalin fixed paraffin embedded (FFPE) slide preparation combined with razor macrodissection, laser capture microdissection (LCM) and the Weke et al. method[19]**

| Comparative analysis of FFPE breast tissue proteomics via In-insert method variants and the Weke et al. method | | | | |
|---|---|---|---|---|
| **Method** | **Weke et al.** | **Macrodissection and In-insert** | **LCM and In-Insert – stained sample** | **LCM and In-Insert – unstained sample** |
| Material | Healthy breast | Healthy breast | Healthy mammary gland acini | Healthy mammary gland acini |
| Sample preservation method | FFPE slide | FFPE slide | FFPE slide | FFPE slide |
| Number of samples processed | 11 slides | 17 voxels | 3 voxels | 2 voxels |
| Area | ~110–150 mm$^2$ entire slide | 1.7–9.9 mm$^2$ voxel | 50 × 50 µm (0.0025 mm$^2$) voxel | 50 × 50 µm (0.0025 mm$^2$) voxel |
| Thickness | 15 µm | 15 µm | 5 µm | 5 µm |
| Estimated cell count | 0.4–5 millions | 20–200 thousands | Up to 50 cells | Up to 50 cells |
| Protein groups (FDR < 1%) | 818–1198 | 301–1184 | 225–448 | 268–358 |
| Peptides (FDR < 1%) | 2744–6149 | 607–5976 | 556–1174 | 763–1221 |
| Detergent and its concentration prior MS | No detergent | 0.03% dodecyl maltoside (DDM) | 0.03% dodecyl maltoside (DDM) | 0.03% dodecyl maltoside (DDM) |
| Staining | No | No | Hematoxylin staining | No |
| Main advantages and disadvantages | Cost effective | Cost effective | – | – |
| | Requires only basic laboratory equipment | Requires only basic laboratory equipment | Availability of LCM | Availability of LCM |
| | No spatial resolution | Spatial resolution up to ~1.7 mm$^2$ FFPE voxel | Excellent spatial resolution starting at 50 × 50 µm, 5-µm thick FFPE voxel | Excellent spatial resolution starting at 50 × 50 µm, 5-µm thick FFPE voxel |
| | Higher protein/peptide loss | Low protein/peptide loss | Low protein/peptide loss | Low protein/peptide loss |
| | Mandatory desalting step | Without desalting step | Without desalting step | Without desalting step |
| | Without robotic platform | Without robotic platform | Without robotic platform, but required LCM technology | Without robotic platform, but required LCM technology |
| | Entire FFPE slides | Macrodissected FFPE voxels | Delicate tissue sub-compartments | Delicate tissue sub-compartments |
| | Hematoxylin staining not applicable | Hematoxylin staining not applicable | Hematoxylin stain could contaminate peptide digests if larger voxels dissected >50 × 50 µm | Hematoxylin staining not applicable |
| | – | Buffer evaporation | Buffer evaporation | Buffer evaporation |
| | – | Complicated voxel transport to the insert | Complicated voxel transport to the insert | Complicated voxel transport to the insert |
| | Entire slide must be used | Less precise macrodissection of larger tissue sub-compartments | Hematoxylin staining indicates LCM dissected area | LCM dissected area of interest has low contrast to surrounding |

Additionally, In-insert processing of LCM microdissected stained an unstained BFPT voxels of same size and cell type processing was benchmarked. The MS-FRAGGER search engine only was used for comparing protein and peptide identifications.

keratins rank at similar positions across A, B, and C plots suggesting that they do not originate from contamination. In addition, similar patterns of keratin ranking suggest similar extraction selectivity of In-insert method applied over voxels compared to the classical methods applied over entire breast cancer FFPE slides. Additionally, we prepared a spectral count heatmap to compare the spectral count of keratins in healthy BFPT from 6 donors, prepared by the method described by Weke et al. with the keratin spectral count in 17 voxels prepared using In-insert method from a single donor's healthy BFPT (Supplementary Fig. 3d). The heatmap reveals that keratins with higher spectral counts in the In-insert processed samples also exhibit higher spectral counts in FFPE tissue slides prepared using the protocol from Weke et al. Furthermore, the heatmap demonstrates a general consensus of low spectral count keratins in compared methods.

**Qualitative proteomic evaluation of the In-insert protocol using mass spectrometry data from macrodissected mammalian breast FFPE tissue slide**

An in-depth qualitative proteomic data analysis further evaluated the effectiveness of the new In-insert proteomic FFPE tissue processing protocol

and generated a spectral library for downstream spatial DIA label-free quantitation. A multi-search engine strategy, including MSFragger, Comet, and MaxQuant proteomic searches of DIA and DDA data, were applied to identify proteins and peptides across 17 BFPT voxels. In total, 11,592 peptides (FDR < 1%) and 5,686 proteins (FDR < 1%) were identified across the entire BFPT slide. Almost 8000 unique peptides (FDR < 1%) (Fig. 2a) and 5,000 proteins (FDR < 1%) (Fig. 2b) identified per average BFPT voxel were grouped to more than 1000 protein groups (FDR < 1%) (Fig. 2c). Qualitative proteomics results suggest that peptides and proteins are common or repeated in BFPT voxels. The peptide and protein identification (Fig. 2a–c and Supplementary Fig. 2c, d) clearly show that the new In-insert proteomics protocol effectively retrieved sufficient proteomic material from each BFPT voxel which was then effectively digested into peptides. Furthermore, Supplementary Fig. 2a, b shows that the peptide digest is sufficient for multiple LC-MS/MS analyses per single macrodissected voxel, allowing the mass spectrometer to acquire reasonable base peak chromatograms (BPCs) or TICs. If the FFPE sample is voxelated into a higher number of voxels (e.g., 100 voxels) requiring longer instrument time, the peptide separation LC gradient could be shortened up to 60 min without

**Fig. 2 | Qualitative analysis of peptides and proteins identified across FFPE tissue voxels. a** Total number of unique peptides identified using a multi-search engine strategy (MaxQuant – "MQ", MSFragger – "FRG", Comet – "CMT") applied to both DDA and DIA data across 17 BFPT voxels. **b** Total number of proteins identified using a multisearch engine strategy (MaxQuant, MSFragger, Comet) applied to both DDA and DIA data across 17 BFPT voxels. **c** Comparison of MaxQuant with matches between runs (DDA_MQ-MBR), MaxQuant (without MBR) (DDA_MQ), MSFragger and Comet search engines applied to both DDA (DDA_FRG and DDA_CMT respectively) and DIA data (DIA_FRG and DIA_CMT respectively) plotted as number of protein groups identified. The data suggest that MaxQuant with MBR function performs better than other search engines. In addition, **c** shows that DDA data yields more identified protein groups compared to DIA data regardless of the search engine used. **d** The Venn diagram shows the overlap of peptides identified in MaxQuant with MBR, MaxQuant (without MBR), MSFragger and Comet. The Venn diagram shows that MaxQuant with MBR function identifies the most peptides while most peptides identified in other search engines are included in the MaxQuant MBR search result. **e** MaxQuant with MBR function identifies most proteins compared to MSFragger and Comet, regardless of the data used. Figure **e** is consistent with **d**, which describes the same result at the peptide level.

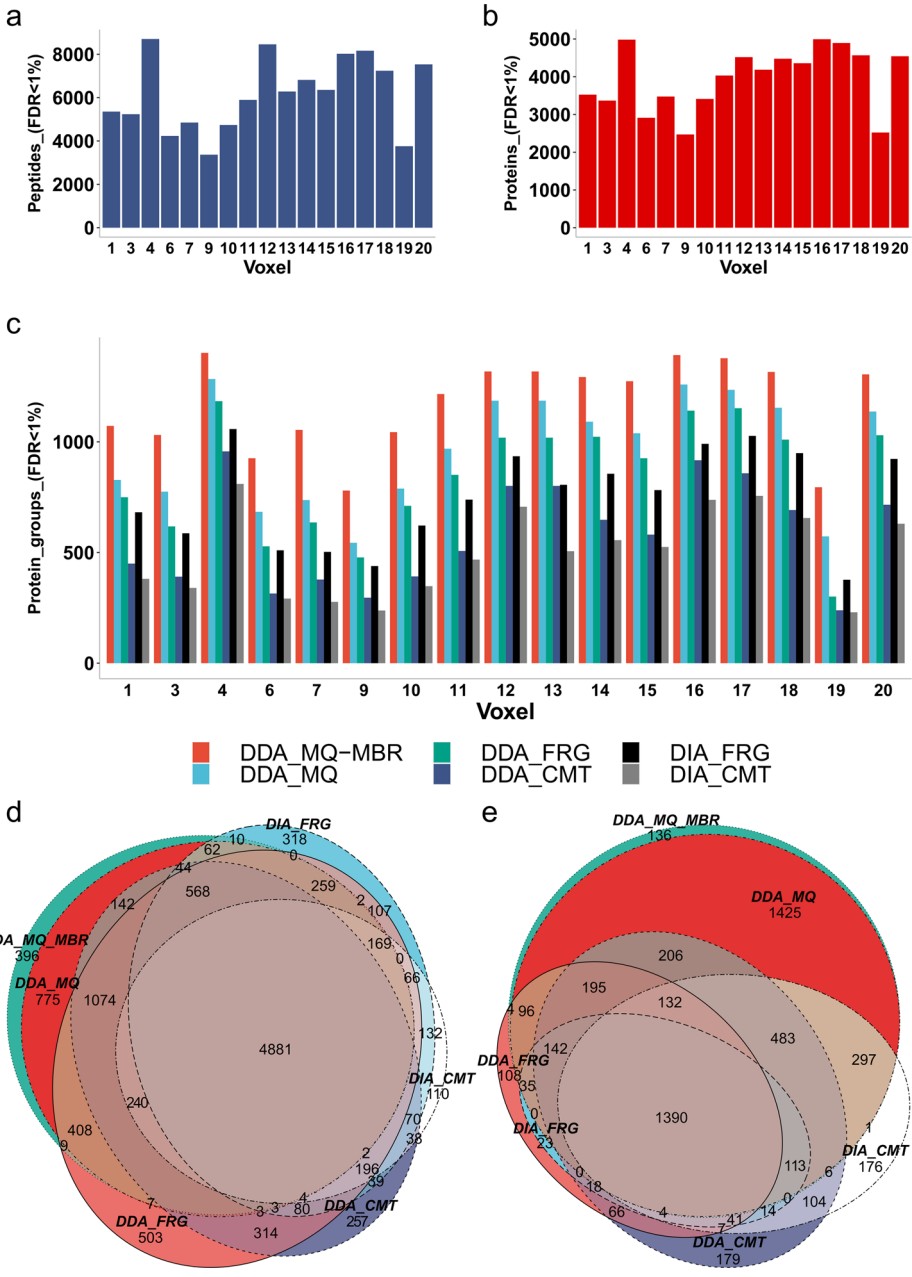

compromising the mass spectrometry identification as demonstrated in the Supplementary Fig. 2e.

Figure 2c shows that the DDA data (red, light blue, green and dark blue columns) provided the most protein groups identified compared with DIA data (black, gray columns) regardless of the search engine used. However, searching the DIA data holds enormous potential as it is only relatively less effective than searching the DDA data (Fig. 2c). Omitting the DDA would save the excess sample needed to subsequently run the DIA, which is essential for label-free quantitative proteomics. Figure 2c (red column) shows that searching the DDA data in MaxQuant with a match-between-runs (MBR) function is the most effective option for searching the data retrieved by In-insert proteomic sample preparation compared with other searches. More than 1000 protein groups were identified in almost all BFPT voxels with MBR MaxQuant function. Including the MaxQuant's MBR function is essential for the voxels yielding compromised protein extracts. Figure 2c shows that the MBR function could increase protein identification by approximately 20% in compromised samples such as BFPT voxels 19, 9, and 7. The data in Fig. 2d show peptide overlap between the compared

search engines applied to both DDA and DIA data. The data in Fig. 2d confirm the observation from Fig. 2c that MaxQuant with the MBR function is the most effective tool for searching the In-insert proteomics data. It also covers most peptides identified by Comet and MSFragger from DDA and DIA data. This observation becomes even clearer when referring to the data in Fig. 2e which show an overlap in identified proteins, where MaxQuant significantly outperforms the Comet and MSFragger search engines. Therefore, Fig. 2c suggests that it is important to search the In-insert DDA data in MaxQuant. However, the data in Fig. 2d, e show that other search engines and a DIA data search might provide several complementary protein identifications that could make the analysis more complete. We highlight that until recently, searching DIA data was uncommon due to missing information about exact precursor m/z information, making it less suitable for identification of proteins compared to DDA. However, DIA data are preferred for accurate protein quantification. Less quantitative DDA methods are better suited for qualitative analyses and thus yield more protein identifications, which is important for building spectral libraries. In contrast, DIA protein quantitation is a sophisticated method that acquires

**Fig. 3 | Subcellular protein localization and trypsin digestion miscleavage. a** Protein subcellular localization retrieved from Uniprot Subcellular Localization. The percentage of proteins in a given localization was determined using keyword analysis. The considered subcellular localizations in the analysis were cytoplasm (red), extracellular space (green), nucleus (dark blue), and plasma membrane (brown). **b** Protein subcellular localization from Gene Ontology (GO). The percentage of proteins in a particular localization was calculated using the same method as in **a**. **c** The percentage of total semi-tryptic peptides across 17 voxels. **d** The percentage of total miscleaved peptides across 17 voxels. **e** The ratio of trypsin cleavage at lysine and arginine residues. The trypsin preference of lysine over arginine at the C-term of identified tryptic peptides shown in blue or at the nearest upstream tryptic peptide (identified peptide N_term minus one aminoacid) shown in red was determined. The analysis shows an overall preference for tryptic cleavage at arginine in almost all voxels.

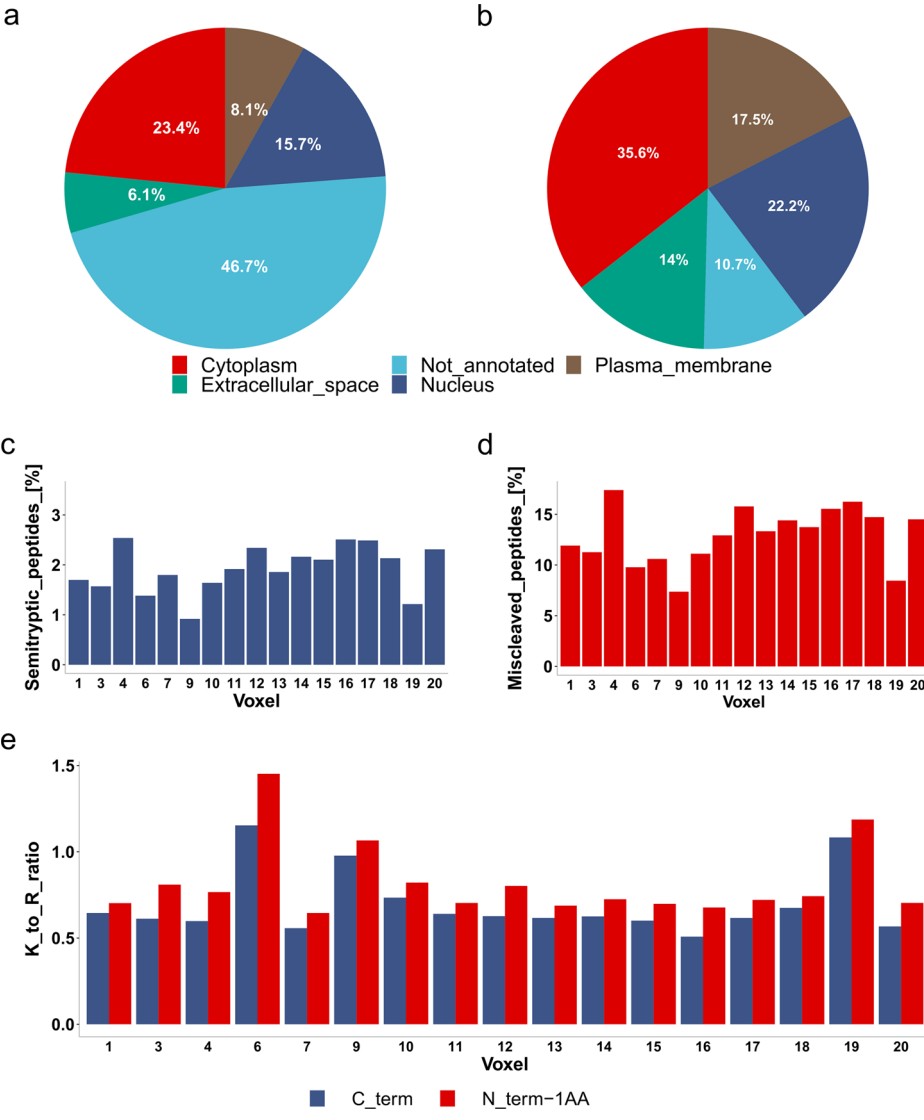

MS/MS spectra in unbiased manner from all peptides. Therefore, we used DIA for protein quantitation in this study.

Protein localization by Uniprot Subcellular Localization (Fig. 3a) and Gene Ontology (GO) Protein Localization Subcellular (Fig. 3b) was performed to reveal protein extraction biases for each tissue/cell region. Almost 5,690 proteins were divided into plasma membrane, nuclear, extracellular space, and cytoplasmic proteins based on Uniprot Subcellular Analysis and GO Protein Localization Subcellular keyword analyses. The data in Fig. 3a, b indicate that the In-insert proteomics workflow effectively extracts proteins from all investigated subcellular locations. A relatively high number of not-annotated proteins were observed in Uniprot Subcellular Analysis compared to GO Protein Localization Subcellular. However, both methods (Fig. 3a, b) are in agreement when comparing protein subcellular localizations. Surprisingly, no biases were observed with respect to the extracellular space proteins (Fig. 3a, b, green) and the plasma membrane proteins (Fig. 3a, b, brown), although steps for mechanical tissue homogenization or the use of a unique lysis buffer system for hydrophobic proteins were not included in the workflow.

It is known that proteins extracted from FFPE tissue are post-translationally modified during fixation[21]. Moreover, protein extraction from FFPE tissue is harsh, leaving space for trypsin miscleavage and possible formation of semi-tryptic peptides. Therefore, semi-tryptic peptides were evaluated in the In-insert prepared BFPT voxels in Fig. 3c. The data in Fig. 3c

show a reproducibly low percentage (up to 2.5%) of semi-tryptic peptides across the BFPT voxels. In addition, a relatively low miscleavage rate of 10–15% was observed across the BFPT voxels (Fig. 3d). The data in Fig. 3e evaluate the lysine to arginine ratio at the tryptic cleavage site either at the C-term of the identified peptide or at the C-term of the neighboring upstream tryptic peptide to the identified peptide. The data in Fig. 3e indicate that in almost all FFPE tissue voxels, tryptic peptides are almost two times more likely to arise from cleavage at an arginine tryptic cleavage site compared to a lysine tryptic cleavage site. Taken together, the data in Fig. 3e suggest that modifications to lysine residues or near to lysine residues may contribute to tryptic miscleavage, if 1:1 distribution of lysine:arginine in the identified proteome is assumed. Therefore, an investigation of the landscape of protein post-translational modifications within In-insert processed BFPT voxels was performed (Fig. 4). A FragPipe open search was used to identify the most common mass shifts (with the exception of common carbamidomethylation, methionine oxidation, and C-term acetylation mass shifts) in BFPT voxels. Surprisingly, only 17.5% of peptides were post-translationally modified. The data in Fig. 4a show the percentage of total peptides modified by the 8 most common mass shifts identified in In-insert processed BFPT voxels. A mass shift of +16 Da was identified on more than 6% of total peptides, +14 Da on almost 3% of total peptides, and +32 Da on almost 3% of total peptides (Fig. 4a). The data in Fig. 4b show the distribution of the identified modifications among the amino acids. Proline is

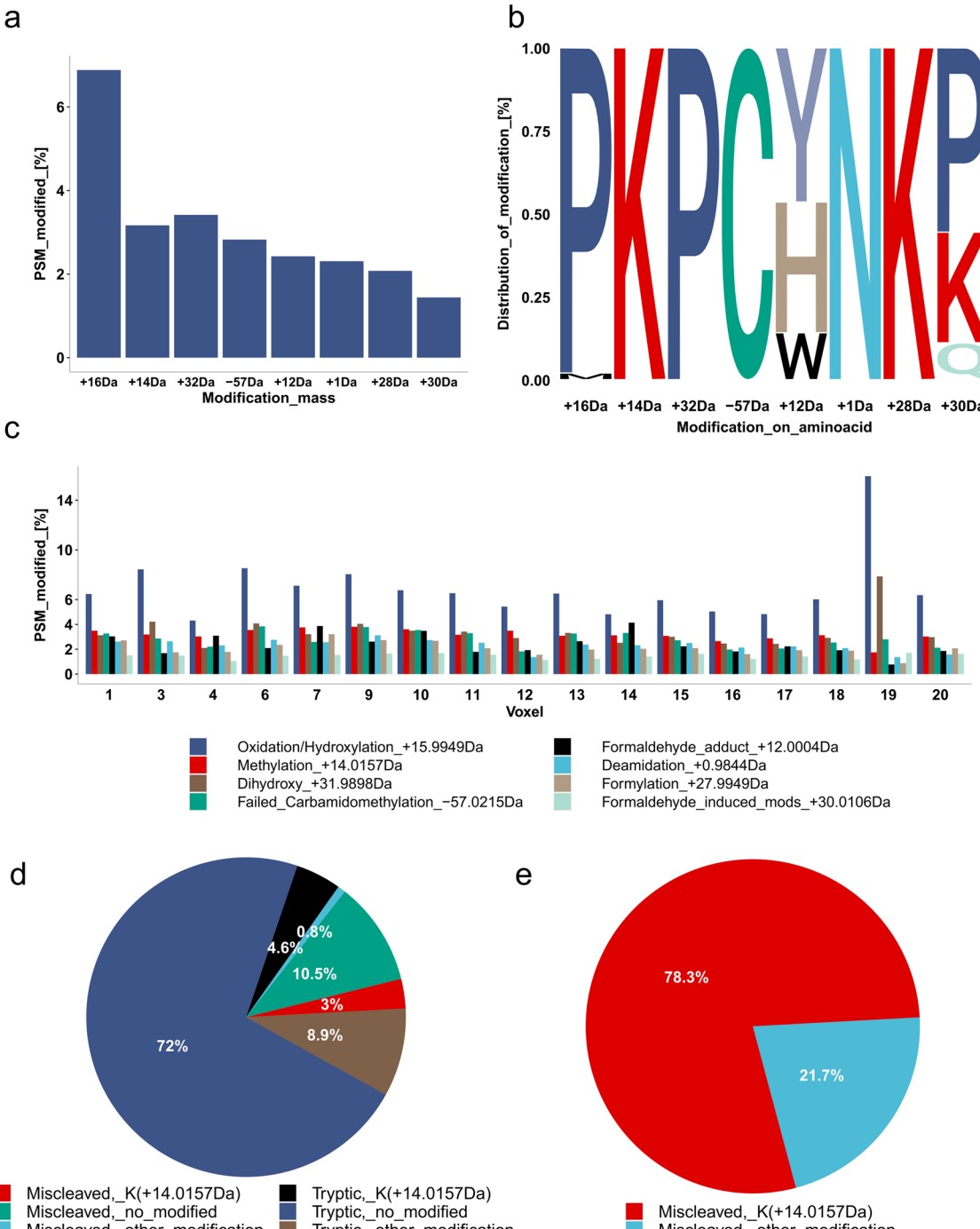

**Fig. 4 | Analysis of modification landscape in In-insert processed FFPE samples.**
**a** Frequency of the 8 most abundant mass shifts expressed as a percentage of the total peptides identified. **b** Analysis of distribution of the 8 most abundant mass shifts among amino acids. The size of the letter abbreviation of the amino acid reflects the percentage of the total modified peptides that have the mass shift on the aminoacid.

**c** Comparison of the frequency of the 8 most abundant mass shifts across 17 voxels expressed as a percentage of the total peptides identified. **d** Analysis of trypsin miscleavage with respect to lysine methylation, expressed as a percentage of total peptides. **e** Analysis of trypsin miscleavage with respect to lysine methylation, expressed as a percentage of modified and miscleaved peptides.

the most frequently modified aminoacid with mass shifts of +16 Da and +32 Da (Fig. 4b). Strikingly, +14 Da and +28 Da modify almost exclusively the lysine residues. Further analysis revealed that the +16 Da mass shift is oxidation/hydroxylation and the +32 Da mass shift is the dihydroxy post-translational modification. The +14 Da mass shift is associated with methylation, and the +28 Da mass shift associated with formylation. The data in Fig. 4c reveal the modification landscape in each voxel, and as expected, the ratio of modification abundance is relatively conserved among BFPT voxels. Proline oxidation and hydroxylation dominate in all voxels

and reproducibly modify up to 6% of the peptides on average, mainly on proline. The methylation adduct (Fig. 4c) affects nearly 3% of the total peptides at their lysine residues (Fig. 4a–c). Taken together, the results from Fig. 4a–c raise the question of whether lysine methylation (K + 14 Da) contributes to the miscleavage of In-insert processed BFPT voxels. The data in Fig. 4d stratify the peptides based on miscleavage relative to lysine methylation. Most peptides are tryptic and not modified, while nearly 11% of peptides are miscleaved and not modified. Interestingly, 3% of the total peptides are miscleaved and modified by methylation, and only less than 1%

are miscleaved and carry modifications other than methylation. Therefore, the data in Fig. 4e were generated to show that a miscleaved and modified peptide is most likely methylated on lysine. In addition, a representative spectrum of the observed phenomenon in a miscleaved peptide IAVAQYSDDVK$_{methylated}$ (+14.0156 Da)VESR, where miscleaved lysine eleven is methylated, was included as Supplementary Fig. 4. Methylation on lysine slightly contributes to miscleavage and should be considered as a variable modification in database search along with other modifications such as Oxidation/hydroxylation and Dihydroxy on proline, as Fig. 4 and Supplementary Fig. 4 suggests.

## Spatial label-free DIA quantitative proteomics of FFPE breast tissue voxels

Qualitative analysis revealed that protein identification across the BFPT slide consistently identified similar protein sets within voxels. Therefore, more sophisticated quantitative DIA mass spectrometry was used to reveal quantitative spatial changes in protein levels between BFPT voxels. DIA cycles within a user-defined precursor mass range split into so-called precursor windows[22]. Therefore, unlike DDA methods, the DIA method provides unbiased MS/MS spectra of all detectable precursor ions entering the mass spectrometer at a given precursor range and time[22,23].

DIA data analysis was performed using only publicly available freeware. Skyline – MSstats quantitative pipeline restricted to unique peptides in the Swissprot *Homo Sapiens* database (01_2022) resulted in the quantitation of 1368 proteins (peakgroup FDR < 1%) across BFPT voxels. Of these, 453 proteins were fully quantitated across investigated voxels and used for downstream analyses. The reproducibility of DIA protein quantitation within BFPT voxels subjected to In-insert processing was evaluated in Supplementary Fig. 5. The results demonstrate favorable correlation coefficients for quantitative data, particularly among certain neighboring voxels, as shown in Supplementary Fig. 5a. Notably, some voxels located at the slide's edge differ more dramatically compared to others, exhibiting lower correlation coefficients. The protein intensity heatmap (Supplementary Fig. 5b) reveals relatively consistent patterns of quantitated protein intensities across voxels. This further confirms observed correlation in quantitative data. To determine variability among voxels, principal component analysis (PCA) analysis was conducted (Supplementary Fig. 5c). PCA plot highlights that voxels with similar protein intensity profiles cluster in the PCA plot, while some of voxels exhibiting distinct protein intensity profiles are positioned farther apart, reflecting relatively higher proteotype dissimilarity.

## A method for screening spatially differentially triggered proteomic pathways in DIA data generated from In-insert processed BFPT voxels

Proteins with the highest standard error (SE) of protein intensity across 17 BFPT voxels were subdivided using a *filter_data* function from the *spatialHeatmap* R package. A cut-off for protein intensity SE (SE > 0.18) was applied to filter the most spatially changed proteins across BFPT voxels. The first insight into the list of most spatially changed proteins across the BFPT slide (Supplementary Table 1) revealed known glucose transporter type 1, erythrocyte/brain (GTR1) and Band 3 anion transport protein (B3AT) erythrocyte related proteins originating from blood that are commonly abundant in tissues. Therefore, the accuracy and biological relevance of a newly established spatial DIA quantitative pipeline was investigated on this blood protein set. The spatial regulation of GTR1 compared to other fully quantitated proteins was calculated (Supplementary data 1) and visualized using an *adj_mod* function and a *network* function implemented in the *spatialHeatmap* R package. The mutual spatial regulation of adjacent proteins was plotted as an adjacency network using the *network* function (Fig. 5a). The most closely spatially regulated proteins to GTR1 were hemoglobin subunit delta (HBD), delta-aminolevulinic acid dehydratase (HEM2), B3AT and spectrin beta chain, erythrocytic (SPTB1), which are well known blood and erythrocyte related proteins. To ensure *adj_mod* and the *network* function

performance, spatial protein intensity heatmaps of GTR1 (Fig. 5b), HBD (Fig. 5c), HEM2 (Fig. 5d) and B3AT (Fig. 5e) across BFPT voxels were plotted using *spatial_hm* function of the *spatialHeatmap* R package. Spatial protein intensity heatmaps (Fig. 5b–e) show similar spatial regulatory trends for the selected proteins, confirming the full functionality of *adj_mod* and the *network* functions. Furthermore, our results are consistent with established blood biochemistry and proteomics[24,25]. Next, the 23 most closely spatially regulated proteins to GTR1 (Fig. 5a) were selected and subjected to Search Tool for the Retrieval of Interacting Genes/Proteins (STRING)[25], which returned an interaction map for these proteins (Supplementary Table 2). As expected, almost all of queries were associated with the term "BTO:0000089 Blood" (FDR = 6.93e−14). The intensities of 19 involved proteins was summed and their summed intensity was plotted across BFPT voxels (Fig. 5f) to spatially search for enrichment of the "BTO:0000089 Blood" term across the voxelated BFPT slide. Further evaluation of the 23 selected proteins via tissue expression (tissues), Subcellular Localization (compartment) and Cellular Component (Gene Ontology (GO)) associated queries with terms such as "BTO:0000132 Blood platelet" (FDR = 9.49e−10), "BTO:0000424 Erythrocyte" (FDR = 4.02e−07), "GOCC:0072562 Blood microparticle" (FDR = 2.07e−13) and "GOCC:0005577 Fibrinogen complex" (FDR = 4.52e−08). GO and Kyoto Encyclopedia of Genes and Genomes (KEGG) pathway enrichment analyses related 23 selected proteins via terms such as "Blood coagulation" (FDR = 0.00036), "GO:0042060_WoundHealing" (FDR = 0.00020), "GO:0042744_H.PeroxideCatabolicProcess" (FDR = 2.18e−08), "hsa04610_Platelet Activation" (FDR = 0.0458) and "hsa04610_Complement and coagulation cascades" (FDR = 0.00097). The revealed terms are closely related to blood proteomics and blood biochemistry, confirming the relevance and functionality of adjacency *network* function, which could a priori reveal functionally linked proteins when applied to In-insert spatial proteomic data[24,25].

Spatial heatmaps of summed protein intensity across BFPT voxels were generated for the enriched biochemical processes and pathways such as "GO:0042060_WoundHealing" and "GO:0042744_H.PeroxideCatabolicProcess" to demonstrate the purpose of spatial proteomic data. The summed protein intensity of the involved proteins was plotted using the *spatial_hm* function for both processes (Fig. 5g, h). Similar spatial trends in pathway activation were observed for both processes, as they are associated with blood and erythrocytes and both are mutually related[26]. Taken together, the combination of the In-insert voxelated FFPE tissue proteomic processing with the spatial DIA quantitative mass spectrometry and the computational pipeline provides results that are consistent with the expected molecular biology.

## PIP protein level reflects estrogen and prolactin activity, which affects keratins that may be involved in cytoskeleton remodeling in certain breast tissue regions

PIP is one of the most spatially dysregulated proteins across the investigated BFPT voxels (Supplementary Table 1). The *adj_mod* function revealed similarly regulated proteins to PIP (Supplementary Table 3) and the *network* function generated a network of proteins with similar spatial regulation trends (Fig. 6a). Strikingly, PIP is closely associated with a network of three keratins; keratin, type I cytoskeletal 9 (K1C9), keratin, type II cytoskeletal 1 (K2C1) and keratin, type I cytoskeletal 10 (K1C10) (Supplementary Table 3, entries highlighted in red). Spatial protein intensity heatmaps for PIP (Fig. 6b), K2C1 (Fig. 6c), and K1C9 (Fig. 6d) were plotted to demonstrate a spatial enrichment of these proteins across BFPT voxels. As expected from adjacency networks from data in Fig. 6a and Supplementary Table 3, the expression pattern across the BFPT voxels is closely related among PIP, K2C1, K1C9 and K1C10, thus a link between keratins and PIP was investigated.

Prolactin hormone has been shown to regulate PIP and to have either activating or attenuating role across multiple keratins[27,28]. Therefore, an adjacency matrix (Supplementary Table 3) was used to reveal additional keratins displaying opposing spatial regulatory effects to PIP, further

**Fig. 5 | Spatial quantitative proteomic analysis of blood related proteins and signaling pathways. a** A protein regulation adjacency network for solute carrier family 2, facilitated glucose transporter member 1 (GTR1) was determined based on the adjacency of its spatial quantitative values to the spatial quantitative values of other quantitated proteins across 17 BFPT voxels. **b–h** show the spatial heatmaps of the protein intensity or the summed protein intensity of the revealed biochemical pathway. Protein intensity or summed protein intensity is plotted as a blue and red shades. The intensity is directly proportional to color shading, from the lowest intensity represented as dark blue to dark red representing the highest intensity. **b** A spatial heatmap of GTR1 intensity across BFPT voxels. **c** A spatial heatmap of hemoglobin subunit delta (HBD) intensity across BFPT voxels. **d** A spatial heatmap of delta-aminolevulinic acid dehydratase (HEM2) intensity across BFPT voxels. **e** A spatial heatmap of band 3 anion transport protein (B3AT) intensity across BFPT voxels. **f** The summed protein intensity of "BTO:0000089_Blood" term members plotted as spatial protein intensity heatmap across 17 BFPT voxels. **g** The summed protein intensity of "GO:0042060_WoundHealing" term members plotted as a spatial protein intensity heatmap across 17 BFPT voxels. **h** The summed protein intensity of "GO:0042744_H.PeroxideCatabolicProcess" term members plotted as spatial protein intensity heatmap across BFPT voxels.

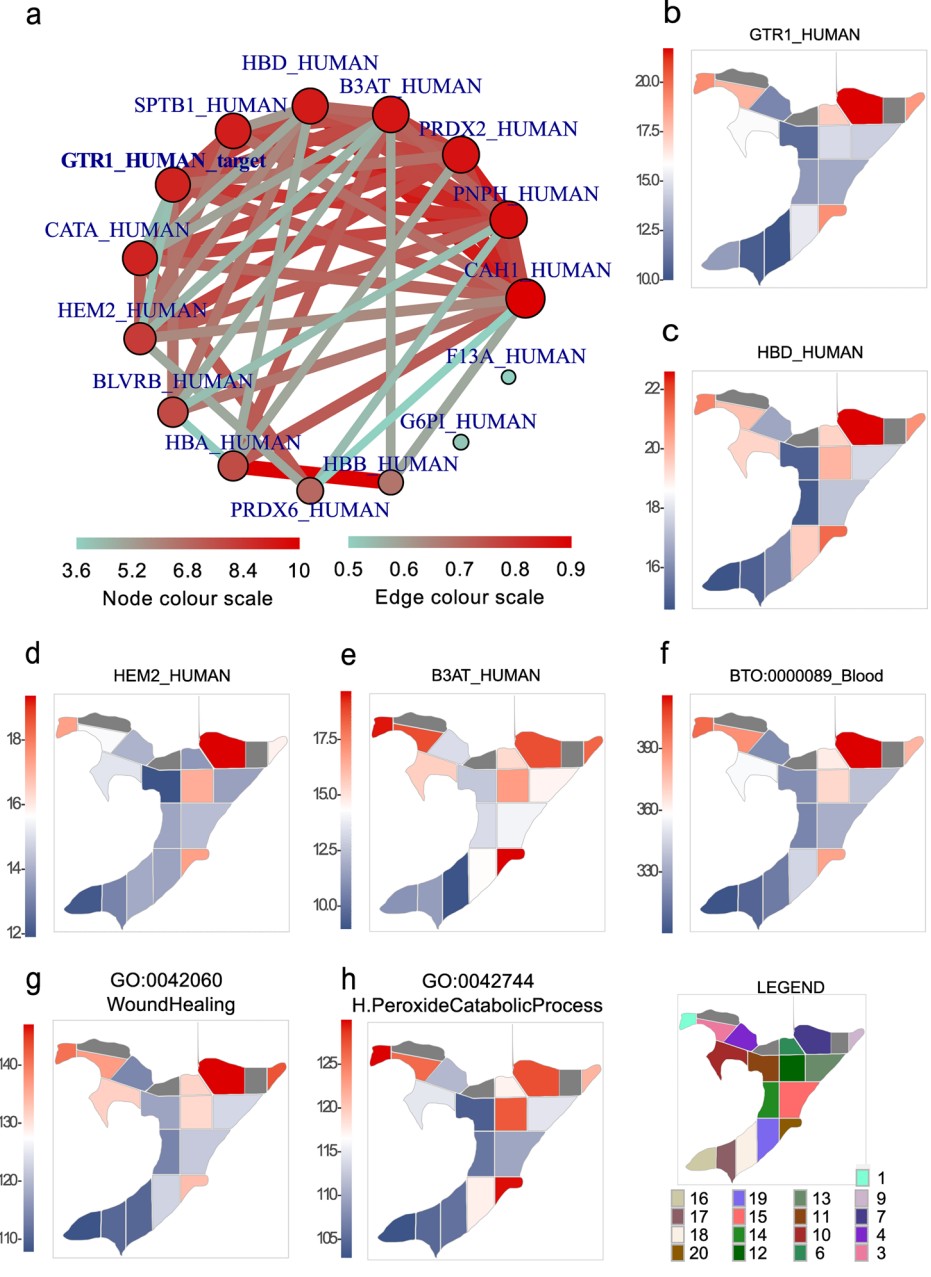

elucidating potential hormonal regulatory events across the voxelated FFPE tissue slide. K1C18 (Fig. 6e), moesin (MOES) (Fig. 6f), both regulated by estrogen, and marginal keratin, type I cytoskeletal 19 (K1C19) (Supplementary Table 3) as well as keratin, type II cytoskeletal 5 (K2C5) (Supplementary Table 3) exhibited an inverse regulation to PIP. Thus, a literature study of the effects of prolactin and estrogen on PIP and keratin levels in conjunction with our observations leads to a speculation that PIP, K1C18 and MOES may spatially reflect estrogen signaling. Alternatively, PIP together with K1C9, K2C1, K1C10, K1C18, K1C19 and K2C5 could spatially reflect the effect of prolactin signaling across the voxels. Hence, a common effect of PIP and keratins was investigated using a STRING protein network association and functional enrichment analysis. A subset of 156 fully quantitated proteins with a cut-off on a spatial regulation adjacency to PIP of less than 5.96e-04 (Supplementary Table 3, entries highlighted in blue) was submitted to STRING. A subset of closely related PIP protein partners displayed a significant enrichment in "GOCC:0045111_IntermediateFilamentCtoskeleton" (FDR = 1.55e−05) (Fig. 6g) (Supplementary Table 4). Figure 6h shows a spatial heatmap of the summed intensity of

"GOCC:0045111_IntermediateFilamentCytoskeleton" proteins across BFPT slide. Notably, prolactin is known to influence cytoskeleton remodeling during mammary epithelial cell differentiation[29].

## Discussion

The FFPE tissue processing protocol in a glass insert reactor termed In-insert FFPE tissue proteomics has been developed. The concept of utilizing a simple container - insert for protein extraction and sample injection into the LC-MS system is not novel. However, our unique approach (Fig. 1) including snap freezing, sonication, tissue homogenization, centrifugation, high-temperature decrosslinking, digestion, possibility of creating a wet chamber and direct sample injection into the LC-MS/MS system operated in DIA mode, has not yet been presented in the context of FFPE spatial proteomics to our knowledge. These distinctive features could be performed without sample transfer rendering the approach novel and well-suited for the field of spatial FFPE proteomics. In-insert protocol does not implement strong detergents, chaotropes, and excessive salts requiring removal prior LC-MS/MS[1,3–7]. Instead, it utilizes nonionic DDM detergent to facilitate

**Fig. 6 | Spatial quantitative proteomic analysis of prolactin-inducible protein (PIP) adjacency network. a** A protein regulation adjacency network of PIP determined from the adjacency of its quantitative values to quantitative values of other quantitated proteins across 17 BFPT voxels. **b–f, h** show spatial heatmaps of protein intensity or enriched term summed protein intensity. The protein intensity or summed protein intensity is plotted as a blue and red shades. The intensity is directly proportional to color shading, from the lowest intensity represented as dark blue to dark red representing the highest intensity. **b** A spatial heatmap of PIP intensity across BFPT voxels. **c** A spatial heatmap of keratin, type II cytoskeletal 1 (K2C1) intensity across BFPT voxels. **d** A spatial heatmap of keratin, type I cytoskeletal 9 (K1C9) intensity across BFPT voxels. **e** A spatial heatmap of keratin, type I cytoskeletal 18 (K1C18) intensity across BFPT voxels. **f** A spatial heatmap of moesin (MOES) intensity across BFPT voxels. **g** The STRING analysis links proteins from PIP adjacency network into the functional protein network of the intermediate filament. **h** The summed protein intensity of the term "GOCC:004511_IntermediateFilamentCytoskeleton" plotted as a spatial protein intensity heatmap across 17 BFPT voxels.

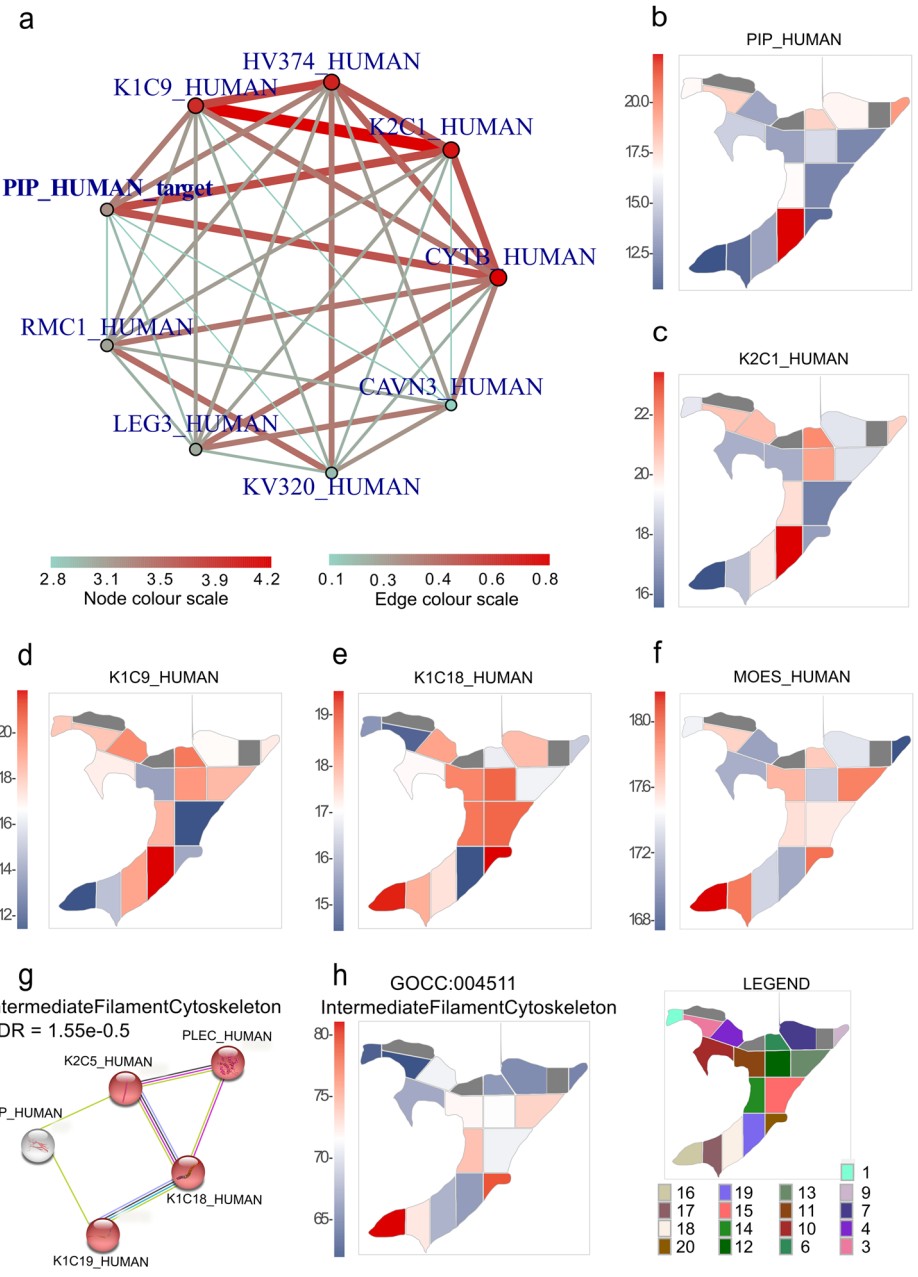

lysis, protein extraction, digestion without interference with LC-MS/MS analysis. Literature review suggested that mass spectrometry compatible concentration of DDM in sample could be approximately 0.02% v/w[17]. Our In-insert proteomics confirms that 0.03% v/w DDM (Table 1, Supplementary Fig. 2) in samples is still mass spectrometry compatible, maintaining microliter scale concentration for direct LC-MS/MS analysis. Moreover, balanced effectiveness of protein extraction from various subcellular localizations was demonstrated when subcellular protein localization was analyzed and benchmarked to preprint by Coscia et al.[30]. Analysis of the protein post-translational modification landscape in BFPT voxels revealed relatively frequent proline oxidation/hydroxylation which is novel in breast FFPE tissue and could be intrinsic modification or introduced during sample processing[31]. In addition, compared with our previous work, a lower trypsin miscleavage rate was observed in In-insert processed FFPE voxels[6].

To evaluate protein extractability and protein yield from healthy FFPE breast tissues important for benchmarking the In-insert method against concurrent methods, we employed a method by Weke et al.[19], identifying approximately 1100 protein groups (FDR < 1%) from 11 entire BFPT slides. Following, we challenged the In-insert sample preparation with LCM dissected and razor macrodissected FFPE healthy breast tissue voxels as a model for a less easily extractable FFPE tissue. With In-insert method, (Table 1) we have identified almost 450 protein groups/1200 peptides (FDR 1%) in LCM dissected 5-µm thick, 0.0025 mm² (50 µm × 50 µm) thick FFPE voxel representing several tens of cells (Supplementary Fig. 1e), while more than 1100 protein groups/6000 peptides/3500 proteins (FDR 1%) from a 15-µm thick, 3 mm² macrodissected breast FFPE tissue voxel. A literature search to compare the In-insert method to intersecting non-robotic FFPE processing protocols demonstrates the In-insert's method effectiveness in minimizing the input material requirements[1–7,9,11]. It reduces the FFPE tissue processing steps by excluding the peptide desalting step while maintaining sensitivity and spatial context. Additionally, In-insert method exploiting principles of LC-MS/MS proteomics could provide greater in-depth proteome coverage and possibly better quantitative accuracy in comparison to mass spectrometry imaging, which on the other hand provides better spatial resolution and is much less laborious. It has been shown that FFPE

processing implementing LC-MS compatible detergents such as Rapigest, PPS, ProteaseMAX could identify up to 270 proteins out of 1 mm$^2$ 4 µm kidney FFPE slide[9]. Further comparison between the In-insert BFPT LCM processing protocol and a one using citric acid antigen retrieval from LCM dissected samples reveals similar effectivity from FFPE tissue areas 20 times larger with the latter method. There might be a slight loss of sensitivity accompanied with the citric acid antigen retrieval protocol due to the desalting step, which in contrary, is omitted in the In-insert protocol[32]. A comparison of FFPE tissue input material between In-insert method and TMT FFPE processing method is challenging because the TMT method involves sample pooling and the use of a carrier proteome to enhance method sensitivity. Generally, TMT method seems more effective than In-insert method, approximately 5600 proteins have been identified from human FFPE substantia nigra tissue using the TMT method and more than 8000 proteins identified and quantified in lung FFPE tissue[13,14]. Nevertheless, In-insert FFPE tissue slide processing in combination with LCM could be performed with sufficient spatial resolution, potentially allowing resolving even distinct regions within a tumor or tissue voxels with several tens of micrometers in lateral size. Our data clearly suggest that the In-insert method with LCM dissection could retrieve a sufficient proteomic sample from FFPE tissue sections smaller than 1 mm$^2$ (Supplementary Fig. 2b–d) which determines the approach for use in the spatial research of FFPE tissue features such as distinct zones of tumor (e.g. tumor infiltrate, necrotic areas, infiltrative front, tumor islets or blood vessels) and distinct adjacent tissue features. Literature search suggests that In-insert method combined with LCM performance matches autoPOTS if number of proteins identified from approximately 50 µm lateral dimensions voxel was compared. It should be noted that the liver tissue voxels processed by autoPOTS which were 10-µm thick might be more easily extractable compared to 5-µm thick normal breast FFPE tissue voxels processed by In-insert method. nanoPOTS method is more effective than In-insert method, extracting almost twice as more protein groups from the FFPE tissue voxel of same size. autoPOTS and nanoPOTS provides excellent sensitivity and spatial resolution, however it must use a robotic platforms[15,16]. On the other hand, In-insert proteomics is more labor intensive, requires manual processing and suffers from excessive buffer evaporation during sample decrosslinking. Its main advantages are versatility, low complexity, manageability, minimal requirements for laboratory equipment and low cost (Table 1) if compared to robotic platforms. Parallel sample processing combined with a 1 h acquisition time allows relatively rapid analysis of up to 20 voxels per 24 h, which is important for large scale studies.

For proper interpretation it is often necessary to correlate such spatial DIA data with histological information. Hematoxylin is a frequently used histological stain. Hematoxylin must be chemically oxidized to develop the color, therefore chemical noise is introduced during staining and it could later adversely impact the MS ionization process[33]. Nevertheless, several LC-MS/MS spatial proteomics researches were successfully performed on sub 0.04 mm$^2$ hematoxylin stained voxels[34,35]. Therefore, we demonstrated processing of both hematoxylin stained and unstained LCM dissected FFPE tissue voxels (0.0025 and 0.01 mm$^2$) processed via In-insert protocol. Our TICs from LC-MS analysis suggest that for the smallest voxels the effect is negligible, however, for larger voxels the adverse effect of hematoxylin staining might potentially compromise LC-MS analysis in terms of affecting ionization efficiency. Introducing chemical contamination and matrix effect together with increasing background noise makes it challenging to detect low abundance peptides. Therefore, we do not recommend In-insert sample processing of hematoxylin stained FFPE tissue in combination with macrodissection. In addition, the distribution of hematoxylin staining is unequal across a tissue slide, thus complications might occasionally emerge even with smaller hematoxylin stained voxels retrieved by LCM.

Biological relevance of the quantitative results retrieved from 17 macrodissected healthy breast slide voxels processed via In-insert protocol was first demonstrated on blood proteomics and biochemistry. Spatial DIA data analysis pipeline a priori linked functionally related proteins involved in vascularization, wound healing processes or blood contamination via

exploiting their spatial regulatory adjacency networks. Moreover, hydrogen peroxide catabolic processes were linked to wound healing based on overlap in their spatial regulation patterns which is in line with the literature[26].

Next, spatial In-insert proteomics functionally linked a protein set consisting of keratin series, plectin (PLEC), MOES and PIP suggesting a common upstream mechanism controlling their protein expression levels. As a prerequisite, we provided evidence demonstrating that the keratins identified in our study are not the result of a contamination but originate from biologically relevant MS signals. PIP is linked to both physiological and malignant conditions in the mammary gland. It is controlled by prolactin, estrogen and androgens[36–42], while both prolactin and estrogen might affect keratin levels[27,28,43–45]. A literature study of the effects of prolactin and estrogen on PIP and keratin levels in conjunction with our observations leads to a conclusion that PIP, K1C18 and MOES may spatially reflect estrogen signaling. Alternatively, PIP together with K2C1, K1C9, K1C10, K1C18, K1C19 and K2C5 could spatially reflect the effect of prolactin signaling across the breast tissue voxels. Interestingly, STRING analysis linked the members of spatially closely or inversely regulated proteins to PIP as a constituents of intermediate filaments that are major cytoskeleton components (PIP, K1C18, K1C19, K2C5, PLEC). The literature indicates that MOES, is a protein required by estrogen to induce cytoskeletal remodeling, while its level could also be affected by prolactin treatment[46–48]. Nevertheless, several reports claim that estrogen could directly interact with the intermediate filament via keratins 8, 18 and 19[49] and even a direct effect of prolactin on cytoskeletal remodeling and motility of breast cancer (BC) cells has been reported[46,47], which is in line with our findings. Therefore, the most spatially dysregulated PIP could be considered the most sensitive spatial protein biomarker for hormonally triggered cytoskeleton remodeling. PIP may prove particularly valuable for screening breast regions where biochemical processes (including cancer) were triggered by hormonal action[50–54]. Our functional study in KM plotter has proven that there might be a link between PIP and BC, as it has shown that high expression of PIP at protein level in BC patients ($N = 126$) leads to significantly lower overall survival ($p = 0.054$) compared to BC patients with low PIP expression within a timeline of 120 months post-diagnosis (Supplementary Fig. 6)[55]. Nevertheless, locally increased PIP levels are linked also to physiological processes, therefore extensive validation studies and additional spatial markers would be required to further elucidate exact roles of PIP in the healthy breast tissue or in breast cancer signaling pathways.

This way, we demonstrated that spatial heatmaps generated from DIA data might spatially reveal changes in sub-cellular structures, biochemical processes, and biomarkers by extracting summed protein intensities across a FFPE tissue. Only a small portion of the spatial DIA data was used to elucidate the PIP protein network. The spatial DIA data could be reused to generate any spatial network of related proteins listed in the spectral library without need to reacquire the data. This approach could substantially accelerate the understanding of the spatial molecular processes involved in oncogenesis and reveal the proteomic architecture of tumors with similar resolution as robotically assisted autoPOTS. Taken together, the method could be used as a cost effective alternative to current methods for research of distinct sections or areas within FFPE slide.

## Methods

### In-insert sample preparation

Sub-microgram FFPE tissue proteomics by In-insert proteomic sample preparation consists of several simple steps that are inspired by well-established "macro-proteomic" protocols such as in-solution, in-gel and Filter Aided Sample Preparation (FASP)[56,57]. In-insert proteomics steps were optimized to omit robotic platforms and compromise sample processing steps keeping the minimum cost while preserving maximum sensitivity.

### Study design

Breast tissue was retrieved from female patient who were admitted at Prince Hamza Hospital and underwent bilateral breast reduction mammoplasty. Breast tissue was confirmed to be pathologically-free. Specimens were

collected under clinical protocols approved by the Internal Review Board committee at Prince Hamza Hospital (No. 9/2019) and in accordance with ethical standards as laid down in the 1964 Declaration of Helsinki and its later amendments. Breast tissue was processed to a 15-μm thick slices and mounted on glass followed by FFPE fixation and paraffin embedding. A tissue slide of an approximately 100 mm² area shown in Supplementary Fig. 1a was used to generate In-insert spatial proteomics data and, 11 additional slides were used for determining an estimate of the expected protein yield and benchmarking the In-insert method to the method published by Weke et al.[19]. An extra 5-μm thick HE stained and non-stained healthy breast tissue slide was used for LCM processing.

FFPE tissue slide intended for macrodissection in combination with In-insert method was macrodissected into 20 FFPE tissue voxels as shown in Supplementary Fig. 1a and 2b by a razor. The aim was to prepare spatially distinct voxels (approximately 2–10 mm²) to effectively resolve the FFPE slide. Voxels number 2, 5, 8 (Supplementary Fig. 1c) were later suspended from the experiment due to suspicion to polymeric contamination originating from Parafilm seal that entered the insert during centrifugation and dipped into the extract. To prevent the Parafilm seal entering the reaction mixture the inserts could be placed into the vials and caps with septum could be used instead to prevent evaporation, alternatively a support from half cut pipette tip (lower part, 200 μl pipette tip) could be made to hold the Parafilm in the opening of the insert. Peptide extract (30 μl) from each voxel was divided into two parts the first (10 μl) to generate a fractions for the spectral library, the second (20 μl) for 2 technical replicates measured in DIA intended for label free protein quantitation and one technical replicate measured in DDA.

### Deparaffinization of FFPE tissue for macrodissection
Breast FFPE tissue mounted on a glass slide was deparaffinized by 2 min incubation in a 100 ml beaker topped up with neat xylene. Next, the tissue was rehydrated in 100%, 75% and 50% ethanol in 100 ml beakers each incubation taking 2 min. Breast FFPE tissue slide was then introduced into a 100 ml beaker topped up with LC-MS water for several seconds to prevent quick evaporation of solvent during voxelating.

### FFPE tissue macrodissection into voxels and FFPE tissue scraping
Breast FFPE tissue mounted on glass slide was macrodissected into 20 voxels by a razor macrodissection. A network of lines on circumference of each voxel was drawn and the voxels were numbered (Supplementary Fig. 1a, b). Next, each voxel was scraped into a stack as shown in Supplementary Fig. 1c. Razor was cleaned by deionized water after scraping each voxel. Each FFPE tissue stack was on-glass slide mixed with 3 μl of 0.2% DDM, 10 mM dithiothreitol (DTT) in water and then carefully inspired into a pipette tip. Tissue with lysis buffer was carefully transferred into a bottom part of glass insert (0.1 ml Micro-insert, Lab Logistics Group GMBH, cat. no: 7401744).

### Laser capture microdissection of FFPE tissue into voxels
Tissue sections for microdissection were prepared according to the protocol provided with microdissection system (Zeiss). Briefly, FFPE blocks were cut into 5 μm thick sections using microtome and placed on membrane coated slide (MembraneSlide 10. PEN, #415190-9041-000, Zeiss). Next, the slides were dried in a drying oven at 56 °C and deparaffinized by immersion in xylene twice for 2 min each and distilled water (6 dips). The slides were stained by incubation in Mayer's Hematoxylin solution for 2 min, rinsed in distilled water for 1 min. Finally, the slides were dehydrated by immersion in a reverse series of ethanol dilutions (70%, 96% and 100%) for several seconds each and air-dried at room temperature.

Microdissection was performed using a PALM MicroBeam Laser Microdissection system (Zeiss), equipped with UV laser for cutting (355 nm) by a board certified pathologist. The slides were visualized using the PALMRobo 4.6 software (Zeiss) and areas containing mammary gland acini were selected. Prior to microdissecting the tissue of interest, the laser parameters were optimized for each employed objective and were as follows:

for the 10x objective (Fluar 10×/0.50 M27, Zeiss) energy = 49 and focus = 79, while for the 20× objective (LD Plan-Neofluar 20×/0.4 Korr M27, Zeiss) energy = 44 and focus = 69. Square tissue areas with lateral dimensions of 50, 100 (both using the 20× objective), and 200 μm (using the 10× objective) were cut to determine the In-insert method effectivity. The microdissected tissue voxels were collected using the Laser Pressure Catapult (LPC) into tubes with an adhesive cap (AdhesiveCap 500, # 415190-9201-000, Zeiss) and stored on ice before further processing.

### Collecting the FFPE tissue from the adhesive cap
The adhesive cap was examined under the light microscope at 10× zoom or with magnifying glass to confirm presence of LCM dissected FFPE tissue voxels Supplementary Fig. 1e. The approximate position was marked from bottom side of the cap. Pipette tip was pointed towards the marked place and LCM dissected tissue was mixed with 3 μl of 0.2% DDM, 10 mM DTT in water followed by three cycles of pipetting up and down. Subsequently, the tissue was carefully inspired into a pipette tip with the lysis buffer and transferred into a bottom part of glass insert (0.1 ml Micro-insert, Lab Logistics Group GMBH, cat. no: 7401744) without puncturing the glue. The adhesive cap was once again checked under microscope to evaluate the success of the transfer.

### FFPE tissue decrosslinking, lysis and protein reduction
Glass inserts with FFPE samples and lysis buffer were 2 times snap-frozen in liquid nitrogen. Inserts were 5 min sonicated in a sonication bath (EMMA D60, EMAG) on ice. Glass inserts were stored at −80 °C. Each glass insert with the sample and lysis buffer was tightly sealed by a Parafilm to prevent sample evaporation, alternatively a homemade gastight rubber cap could be used or the best alternative is a vial cap with septum. A wet chamber was made by pipetting 200 μl of deionized water into a 2 ml Eppendorf tube. The sealed insert with the sample and lysis buffer was transferred inside the wet chamber and the cap was tightly closed. The chamber was transferred into a thermomixer (Grant Instruments) and the FFPE tissue sample was decrosslinked for 1 h at 95 °C and 600 RPM. Each insert was checked for any excessive evaporation every 10 min. The LC-MS water was topped up to 3–6 μl and insert was properly resealed if the lysate evaporated excessively.

### Protein alkylation and digestion
Protein alkylation was performed by adding 50 mM iodoacetamide (IAA) in LC-MS water to 10 mM final concentration in the reaction mixture. Protein alkylation was held at room temperature in a darkness for 20 min. Sample was diluted by adding 27 μl of 25 mM ammonium bicarbonate (ABC), 0.74 nmol μl⁻¹ Sequencing Grade Modified Trypsin (Promega) in LC-MS water and reaction mixture was homogenized by pipetting up and down. Inserts were properly sealed by Parafilm and placed back to wet chambers. Wet chambers were placed into the thermostat and proteins were digested overnight at 37 °C. Inserts seals were checked for an excessive evaporation after one hour. Samples were topped by LC-MS water up to 30 μl and properly resealed if evaporation was spotted. Next day Parafilm seals were removed, and the level of liquid was inspected. The inserts were again topped up to 30 μl with LC-MS water in a case of excessive overnight evaporation. For tryptic digestion of LCM samples, the total volume of the digestion mixture was reduced to 15 μl, while keeping the other steps unchanged.

### Sample preparation of benchmarking dataset for estimating protein yield
Eleven normal breast FFPE tissue slides were prepared following the protocol from Weke et al.[19] for estimating protein yield and benchmarking purposes. Briefly, the extraction buffer, consisting of 30% AcN and 100 mM ABC, was added to the samples. Sample extraction was held at 95 °C for 90 min. To reduce disulphide bonds, DTT at a concentration of 700 mM was used, followed by alkylation of reaction mixture with 700 mM IAA for

next 30 min at 37 °C in darkness. The samples were then topped up with 880 µl of water and 120 µl of ABC. Sequencing Grade Modified Trypsin (Promega) was added in a ratio of 5 ng mm$^{-2}$ and digestion was held overnight at 37 °C. Next day, samples were desalted on the Micro Spin C18 columns (Harvard Apparatus) according to manufacturer guidelines and dried on SpeedVac (Eppendorf) from remaining solvents. Data analysis and acquisition was done using the same MS methods and software as described for FFPE voxels.

### Peptide pre-fractionation for spectral library
Approximately 10 µl of each sample were pooled to prepare peptide fractions for a spectral library. The Pierce™ High pH Reversed-Phase Peptide Fractionation Kit (Thermo Scientific) was used to fractionate pooled sample into 8 fractions according to the manufacturer guidelines. Fractions were evaporated in SpeedVac and were resuspended in 20 µl of 0.08% TFA, 2.5% AcN in LC-MS water prior LC-MS/MS analysis.

### Liquid chromatography and mass spectrometry of In-insert proteomic samples
Trifluoroacetic acid (TFA) was added to inserts up to 0.08% (v/v) and LC-MS acetonitrile (AcN) up to 2.5% (v/v). Optionally, each sample could be spiked with iRT retention time standard (Biognosys) according to the manufacturer guidelines. Following, inserts with peptide digests were placed into clean 2 ml tubes without Parafilm sealing at the top. Inserts were centrifuged at 10,000 g/30 min/20 °C. Inserts were placed into LC-MS vials (cat. no: 702 282, Vial N9 11.6 × 32 mm) and introduced into the UltiMate™ 3000 RSLCnano System (Thermo Scientific). Six microliters of the sample were injected and concentrated on an Acclaim™ PepMap™ 100, 5 um particle size, 1 mm inner diameter, 5 mm length C18 pre-column (cat. no: 16045, Thermo Scientific). Injected sample was desalted on precolumn by a 0.08% TFA, 2.5% AcN in LC-MS water at constant mobile phase flow of 5 µl min$^{-1}$ for 10 min. Peptide trapping and desalting was done in "reverse flush" precolumn arrangement. Next, peptides were eluted from pre-column to PepMap™ 100, 2 µm particle size, 1 mm inner diameter, 5 mm length C18 analytical column (cat. no: 164534, Thermo Scientific) by a linear gradient of mobile phase B (0.1% formic acid (FA) in ACN (v/v)) in a mobile phase A (0.1% FA in water (v/v)). Analytical peptide separation has been started at 2.5% B linearly increasing up to 40% B in 90 min (60 and 20 min for gradient length comparison) with a constant flow of 300 nl min$^{-1}$. A column flush at 95% B was performed over next 8 min, and finally 8 min column equilibration at 2.5% B followed. Separated sample was ionized in a nanoelectrospray ion source and peptide ions were introduced into an Orbitrap Exploris™ 480 Mass Spectrometer (Thermo Scientific).

A single replicate of DDA data was acquired by a DDA method composed from full scan and MS/MS scan executed on Orbitrap Exploris™ 480 Mass Spectrometer. The full scan was operated in a profile mode with 120,000 resolution. A precursor range was set from m/z 350 Th to m/z 1200 Th. Normalized AGC target was set to 300% with auto setting on maximum injection time. Each MS scan was followed by a fragmentation of the top 15 most intense precursor ions and acquisition of their MS/MS spectra. The dynamic mass exclusion was set to 20 sec after the first precursor ion fragmentation. Precursor isotopologues were excluded, and precursor exclusion mass tolerance was set to 10 ppm. Minimum precursor ion intensity was set to 5.0e3, and only precursor charge states of +2 to +6 were included in the experiment. The precursor isolation window was set to 2 Th. Normalized collision energy type with fixed collision energy mode was selected. The collision energy was set to 30%. Orbitrap resolution was set to 15,000. Normalized AGC target was set to Standard with 40 msec maximum injection time, and the data type was centroid.

### Data independent acquisition
Two DIA technical replicates of each voxel were measured. LC separation parameters were kept identical to DDA during DIA. Orbitrap Exploris 480 mass spectrometer operated in positive polarity DIA mode accompanied by a full scan at 60,000 resolution. The full-scan mass range was set from m/z

350 Th up to m/z 1450 Th, and the normalized AGC target was set to 300% with 100 msec maximum injection time. DIA method covered a mass range from m/z 350 Th up to m/z 1100 Th with 12 Th window width and 1 Th overlap. One DIA cycle consisted of 62 precursor windows/scan events. Normalized collision energy type with fixed collision energy mode was selected. The collision energy was set to 30% and orbitrap resolution to 30,000. Normalized AGC target was set to 1000% with automatic setting of maximum injection time. Data type was profile.

### Spectral library generation and MS/MS database searches
A multi-search engine strategy was developed to identify In-insert generated peptides and proteins in both DDA unfractionated and fractionated and DIA data. MSFragger 3.4[58] search engine embedded in FragPipe (v.15), Comet (Release 2020.01, revision 2)[59] embedded in TPP v6.0.0. OmegaBlock[60] and MaxQuant 2.1.0.0[61] search engines were used. First, raw DDA files were centroided and converted into mzML and mzXML formats using MSConvert version: 3.0.19094[62]. Raw DIA files were processed to pseudo-DDA data via DIA-Umpire available in FragPipe (v.15)[63]. MS1 extraction mass accuracy was set to 10 ppm and MS2 to 20 ppm. One missed scan was allowed. Other settings were left default for Orbitrap data. Following, extracted pseudo-DDA data were converted to centroided mzML and mzXML format. mzML files were searched using MSFragger and mzXML files using Comet, while raw DDA data were searched in MaxQuant. All searches were done against *Homo sapiens* SwissProt+UniProt search database (01_2022). concatenated with a reverse decoy database containing equal amount of reversed target sequences and common contaminant protein sequences. The closed search was done in MSFragger with + and -8 ppm precursor mass tolerance and with 10 ppm fragment mass tolerance. Enzyme digestion was set to trypsin and two missed cleavages were allowed. Carbamidomethylation of cysteine was set as fixed modification. Variable modifications were set to methionine oxidation, protein N-term acetylation, and methylation of lysine. Data were mass recalibrated, and automatic parameter optimization setting was used to tune fragment mass tolerance. Output file format was set to pep.XML. Precursor mass tolerance was set to 8 ppm and fragment mass tolerance to 8 ppm in Comet closed search. The rest of the settings were identical to MSFragger. MaxQuant main search peptide tolerance was set to 8 ppm during the closed search while the other settings were set identical to MSFragger and Comet or kept default. Searching the formaldehyde induced FFPE tissue modifications was done by open search in MSFragger 3.4. All open search settings were left default. Output form MSFragger open search refined in Crystal-C[64] and further processed by PTM-Shepherd, both implemented in FragPipe (v.15)[65]. Crystal-C and PTM-Shepherd settings were left default and open search result was directly processed.

The MSFragger and Comet closed search results (pep.XML files) were joined and the peptide probabilities were recalculated in PeptideProphet[66] and iProphet[67] which are running as a part of the TPP v6.0.0. OmegaBlock. MaxQuant closed search results (MSMS file) were used further without peptide probability recalculation as it is automatically done within MaxQuant. Recalculated pep.XML and MSMS MaxQuant identification files from both fractionated and non-fractionated DDA files were imported into Skyline-daily (64-bit, 20.1.9.234) and transformed to a spectral library .blib file[68]. The conversion of search files to .blib spectral library was done as follows: The "Score Threshold" was left default (0.99 for PeptideProphet probability or 0.05 for MaxQuant PEP score), only the peptides with better score than threshold score were considered for the spectral library. The FASTA file that was previously used as a search library was used as a background proteome to which filtered peptides mapped. In the library window an "Associate proteins" option followed by selecting "Add all" triggered a script selecting unique (proteotypic) peptide spectra from pep.XML files and MaxQuant MSMS search file, that are overlapping with background proteome. Finally, selected peptides were associated to proteins after finishing the peptide import and the spectral library was stored as .blib file.

## Spatial label-free protein quantitation using DIA data

Spatial protein intensity extraction from DIA data was performed using Skyline-daily (64-bit, 20.1.9.234). Two missed cleavages were allowed in Peptide settings tab. Maximum peptide length was set from 4 to 200 amino acids and no N-term aminoacids were excluded. No modifications were considered. The newly created spectral library was selected in the "Library" tab. Plus 1, to +6 precursor ions and y and b product ions with +1 and +2 charges were included in the experiment. Product ion selection was set from ion 4 to the last ion. The "Auto select all matching transition" option was activated. Ion match tolerance was set to 0.05 m/z and only peptides that have at least 3 product ions were kept in analysis. In addition, if more product ions were available only 6 most intense were kept. DIA option was selected and Orbitrap was set as a mass analyzer. DIA isolation scheme was imported from raw DIA files. Mass analyzer resolution was set to 30,000 at 200 m/z. Only scans within 5 min of MS/MS IDs were considered. Empty proteins were removed from the document. Equal number of reverse decoy sequences were added to targets. mProphet peak scoring model was trained on decoys and targets after DIA files were imported[69]. Extracted quantitative data report was set to include all dependencies (specified in documentation) required for *MSstats 4.0.1.* later used in downstream analysis[70].

## Spatial protein quantitation in MSstats statistical module

Statistical analysis of Skyline extracted DIA data was performed in R (version 4.0.0) package *MSstats 4.0.1.* mProphet q-value threshold was set to q-value < 0.01 to filter out potentially false positive peakgroups. *SkylinetoMSstatsFormat* function was set to keep proteins with one feature and to transform Skyline output to MSstats input. Peptide intensities were log2 transformed and quantile normalized. Protein quantitation across voxels was performed pairwise via mixed-effect models implemented in *MSstats groupComparison* function. p-values were adjusted using the Benjamini-Hochberg method and protein intensity result matrix was exported for downstream analyses. Full protein comparison matrix after statistical significance evaluation across all possible comparisons of proteomes from 17 voxels is available in PXD037609 dataset as described in data availability section.

## Protein filtering and spatial protein intensity plotting

A functionalities of *spatialHeatmap 2.3.0* R package running in R (version 4.0.0) were used to find the most spatially dysregulated proteins and plot the spatial heatmaps and protein spatial regulatory relation networks[71]. Protein intensity matrix was formatted and submitted to *filter_data* function of *spatialHeatmap* R package. A cut-off on summed protein intensity standard error (SE > 0.18) was set to filter the most spatially changed proteins across the BFPT voxels. *adj_mod* function of *spatialHeatmap* R package was used to calculate adjacency matrix of spatial protein regulation among the proteins. Graphical presentation of protein regulation adjacency was plotted via *network* function of *spatialHeatmap* R package relying on output matrix of *adj_mod* function. Spatial protein intensity heatmaps were plotted using *spatial_hm* function of *spatialHeatmap* R package.

## Plotting the qualitative proteomic data

Stacked bar plots, bar plots and pie graphs were generated in *plyr 1.8.6*[72]. and *ggplot2 3.3.6*[73] R packages. Venn diagrams were generated in *Eulerr 6.1.1*[74] package. *ggseqlogo 0.1*[75] R package was used to generate a logo showing the modification frequency at particular aminoacids. Ggpubr 0.4.0 package was used to stack multiple graphs per page. Circlize 0.4.15[76] was used to generate custom color palette for publication. Inkcape 1.2 and Gimp 2.10.32 were used to process the graphics to final panel plots and to generate svg image of breast FFPE tissue slide later used in *spatialHeatmap* R package.

## Reporting summary

Further information on research design is available in the Nature Portfolio Reporting Summary linked to this article.

## Data availability

Mass spectrometry data including raw data files, search results and full protein quantitation results have been deposited to ProteomeXchange with dataset identifier: PXD037609 under username: reviewer pxd037609@e-bi.ac.uk and password: fq17pAeG and in an additional ProteomeXchange repository with dataset identifier: PXD051706 under username: reviewer_pxd051706@ebi.ac.uk and password: ZxVQYDzs[77]. Numerical source data underlying all plots in the manuscript can be found in Supplementary data 2 file. All other data are available from the corresponding authors upon request.

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

## Acknowledgements
This work was supported by the University of Gdansk and International Centre for Cancer Vaccine Science project carried out within the International Research Agendas program of the Foundation for Polish Science, cofinanced by the European Union under the European Regional Development Fund (project MAB/2017/3). We are grateful to Dr. Sofian Al Shboul and Dr. Tareq Saleh from Department of Pharmacology and Public Health, Faculty of Medicine at The Hashemite University, Zarqua, Jordan for providing the formalin-fixed paraffin embedded breast tissue mounted on a glass slide. The authors would also like to thank the CI-TASK, Gdansk, and PL-Grid Infrastructure, Poland, for providing their hardware and software resources.

## Author contributions
J.F. supervised spatial DIA-MS analyses, analyzed spatial DIA-MS data, coordinated the study, and wrote the paper. S.K., T.R.H., and N.M.T. supervised and mentored the study and assisted with revision of the paper. M.B. Performed the laser capture microdissection, helped with the interpretation of the data and assisted with revisions of the paper.

## Competing interests
The authors declare no competing interests.
