## [Peer review file · Communications Biology]

Reviewers' comments:

Reviewer #1 (Remarks to the Author):

The manuscript by J. Faktor et.al describes a novel non-robotic In-insert processing method for the proteomic analysis of formalin-fixed paraffin-embedded (FFPE) breast tissue. The novelty consist in processing FFPE sample voxels in a glass insert by transferring the steps of classical protocols to the microliter scale for a direct LC-MS/MS analysis. The advantages of this approach rely in reducing the FFPE tissue processing steps and cost while maintaining maximum sensitivity and spatial content in an user-friendly manner. Combining this application with a spatial quantitative data-independent mass spectrometry, the authors revealed a correlation of a keratin series and moesin (MOES) with prolactin-induced protein (PIP) indicating their prolactin and/or estrogen regulation. The authors suggest PIP as a spatial biomarker of hormonally triggered cytoskeletal remodeling possibly useful for screening hormonally triggered carcinogenesis hotspots.

This is an interesting approach that could be fruitfully integrated with the available proteomics methods in oncogenesis. Here are my comments:

Major:

1. My main concern is that the authors did not compare this novel "In-insert FFPE" proteomics approach with a basic proteomic approach. Although a reference has been reported for an implemented mass spectrometry method for protein recovery from FFPE samples, such reference describes a method applied on kidney and not on breast tissue. For instance, I would have like to see a comparison of the results obtained from "In-insert FFPE proteomics" with those obtained from a basic proteomic approach on a consecutive slide of the same mammalian breast FFPE sample.

Minor:

2. "Introduction" section:

In the sentence "This problem has partly been overcome by implementing mass spectrometry compatible detergents such as Rapigest, PPS silent surfactant, ProteaseMax or by using direct trypsinization", please include some references related at this matter. Here are some examples: PMID: PMC7900714, 26690073, 21280217

3. "Introduction" section:

"Implementing TMT labeling of FFPE subsections, followed by pooling and adding a booster channel substantially improves sensitivity of protein detection", please define "TMT"

4. "Results" section:

"FIG1 demonstrates the versatility of the glass insert, which allows repeated snap-freezing in liquid nitrogen, bath sonication, centrifugation and de-crosslinking at nearly 100°C followed by protease digestion at microliter scale." None of the steps "repeated snap-freezing in liquid nitrogen, bath sonication, centrifugation" are indicated in Figure 1., please show or include those steps in the workflow of Figure 1.

5. "Materials and methods", "Study design" subsection:

In the sentence "Voxels number 2, 5, 8 (Supplementary file 1C) were later suspended from the experiment due to suspicion to polymeric contamination." It is very hard to see the number of the voxels in the supplementary file 1 B, C. This reviewer suggests numbering every voxel digitally.

6. Supplementary Figure 2D: Why did the authors choose the 30x and not the 10X diluted sample to test the effect of gradient length on protein identification? In the 10x diluted sample a higher number of peptides was identified as showed in Figure 2C.

7. Figure 3C, D, In the caption it is stated that the semi-tryptic peptides and the miscleaved peptides are considered across 17 voxels, but the figure shows 20 voxels, please clarify.

8. Figure 3E. Do the authors know why only voxel number 6 and 19 showed a preference for tryptic cleavage at lysine? Did these voxels follow the same exactly sample preparation procedure of the other voxels?

9. Figure 6. The authors stated that type I cytoskeletal 10 (K1C10) protein is closely associated to PIP, however K1C10 is neither represented in the network of Figure 6A nor as a spatial protein intensity

heatmap like the other keratins. On the other hand, there is a discrepancy with the caption of Figure 6, where it is stated that panel E is the spatial heatmap of K1C10, but in the figure it is represented by the spatial heatmap of K1C18. Please correct.

10. "Discussion" section: "PIP may prove particularly valuable for screening breast tumor regions..."

This result comes from the analysis of only one sample of breast cancer, the authors should discuss that this result needs to be confirmed in a larger number of samples.

Reviewer #2 (Remarks to the Author):

This study presents a novel approach for proteomic analysis of small areas of FFPE tissue. The method presented has a number of advantages, primarily focused on cost-effectiveness and accessibility. It is compatible with small regions of human FFPE tissue, which is of high relevance to the broad medical research community. However, there are a number of weaknesses of the study that need to be addressed prior to publication, which I have listed below:

1. I recommend rephrasing the justification for this study in the introduction. I agree that a method that does not rely on expensive equipment or reagents is advantageous. However, there are multiple statements that suggest this new method is better than these other approaches, which isn't the case (given the relatively small number of protein group identifications for the amount of tissue analyzed). I suggest that it would be better to focus the justification for this new method on the need for a simplified sample preparation approach that is comparable to current approaches, as this is still an advance for the field. I suggest rephrasing paragraph 2 of the introduction to account for this as how this paragraph is currently presented raises many questions such as: why is this method better for spatial proteomics than mass spectrometry imaging? Why is it better to use macrodissection rather than more precise microdissection using LCM? Can this method detect more protein groups than the others they are comparing to using the same amount of tissue? It becomes apparent further in manuscript that the proposed method is not better than previous ones, but instead has its own unique advantages while maintaining somewhat comparable protein group identifications, specifically with regards to accessibility and cost effectiveness (but not necessarily the proteomic data it can generate).
2. The "spatial resolution" is quite low using this approach (based on regions that are between 2-10mm²). Given this, I would temper the references to the advantages of this approach for "spatial proteomics" as each of these regions would contain thousands of cells and other spatial proteomics approaches have a much higher resolution (eg mass spec imaging). I suggest that the authors acknowledge this as a limitation, but highlight instances where such an approach would be useful (e.g. tumor vs surrounding tissue perhaps).
3. The area of tissue that they are using to demonstrate their approach is quite large (2-10mm² regions from 15um thick sections) in comparison to other similar approaches using FFPE tissue (often regions half this size and sections <10um). Given this, it is difficult to determine direct evidence that their method generating better (or even comparable) results given that there are multiple other approaches in the literature capable of identifying similar/more protein groups using smaller amounts of FFPE tissue. Please remove/rephrase the statements suggesting their method is better than others that use a much smaller amount of starting tissue as these are not an equivalent comparison and do not support the superior nature of their approach (e.g. Intro: "Nevertheless, implementation of these protocols on 1mm² FFPE tissue led to identification of only several hundred proteins" and Discussion: "It has been shown that FFPE processing implementing MS-compatible detergents such as Rapigest, PPS, ProteaseMAX could identify up to 270 proteins out of 1 mm² 4 μm kidney FFPE slide"). Furthermore, I don't agree that serial dilution after sample preparation is an equivalent comparison to previous studies that have analysed smaller amounts of tissue – one of the main roadblocks with these types of studies is sample loss when processing microscopic amounts of tissue – diluting a larger amount of starting tissue post-processing does not address this issue and is therefore cannot be directly compared to other approaches that use a smaller amount of starting tissue. A better comparison would be analysis of the same size regions of tissue using their approach. If the authors

wish to show evidence that their method works using a smaller area of tissue, then this must be done using a smaller area of starting tissue.

4. I suggest including a comparison of their results to the known proteome of breast tissue (which could either be obtained from direct analysis of the same frozen tissue or more broadly from a comparative dataset in the literature that provides a more comprehensive analysis of breast tissue). This would be useful to see whether the proteins that are detected in this study are the most abundant ones in breast tissue or if there is an overabundance of common contaminants (as is often the case when analysing small regions of FFPE tissue). This is important as many of the highlighted proteins in this study are keratins, which are also known contaminants – it would be useful to show that this is expected in breast tissue.

5. I recommend a clearer explanation of the advantage of in-insert analysis; this is currently unclear and is presented as a defining advantage of this protocol.

6. No complete dataset of results is provided in supplementary tables – only the top 11 most abundant proteins is included in Sup3. Please include a supplementary table of results showing to relative abundance of all protein groups between voxels.

Reviewer #3 (Remarks to the Author):

In the current study, Faktor and colleagues develop a non-robotic In-insert FFPE proteomics approach, which allows spatially resolved quantitative proteomics analysis of FFPE tissue. The In-insert proteomics approach is applied to the sample preparation of 2-10 mm² 15 µm thick FFPE breast tissue voxels with spatial quantitative data-independent mass spectrometry. However, important details are missing in this paper. Additionally, the authors tend to misleadingly over-represent their work as a non-robotic user-friendly or effective method superior to other methods. But in many aspects, it is not superior to some of the methodologies discussed in the paper.

Major comments:

1. The workflow in Figure 1 showed us the authors aim to develop a non-robotic In-insert FFPE proteomics approach to allow spatially resolved quantitative proteomics analysis of FFPE tissue in a user-friendly and economical manner. However, it seems to not a user-friendly or effective method, for it includes cumbersome procedure involving many manual processing steps, which is not user-friendly and may exist potential sample loss. In L30 the authors state that in other sample preparation methods “the detergents and chaotropes must be removed from the sample prior digestion or LC-MS analysis which often leads to excessive sample losses”, but more obvious sample loss may exist when the manipulator scraping each voxel and transfer tissue and lysis buffer into glass insert by using the In-insert FFPE proteomics approach.

2. L44 the authors indicate the In-insert FFPE proteomics approach is a “novel, effective, user-friendly and economical ” method, but the sample preparation procedure seems to be a simple combination of other publications discussed in this paper, from which we could not see the novelty of this method. In Figure 2, the authors benchmark their approach, an important analysis that is missing here is about the quantitative reproducibility of their approach. Besides, the authors also need to provide the information about the area of each voxel and the corresponding protein identification depth, and how does this compare to other sample preparation protocols ?

3. L95-101 and Supplementary file 2C the authors state “LC-MS/MS analyses of 30x, 90x, 300x and 1800x diluted samples suggest that the method also works for less than 1 mm² and thinner FFPE voxels”. Simply dilute the sample may not support the statement that the method can work for less than 1 mm² and thinner FFPE voxels. Directly process sample that less than 1 mm² may better support this statement and it is better for the authors to give us the information about the minimum area which can be processed by the In-insert FFPE proteomics approach.

4. In Figure 2C, the authors do not give enough explanation about why the DDA data identified more protein groups than DIA regardless of the search engine used, for DIA will offer superior depth in most cases. The detail information about the spectrum library built in this study for DIA is also missed in this paper.

5. In Figure 2, the authors compare DDA and DIA use different engines and draw the conclusion that MaxQuant significantly outperforms the Comet and MSFragger (L117). The authors also find DDA data in MaxQuant with a MBR function is the most effective option for In-insert proteomics sample preparation compared with other searches (L107-109). But why the authors finally choose DIA combine with Skyline for further analysis which may be confusing for readers.

Minor comments:

1. The workflow introduced in Figure 1 could not well depict your experimental procedure. For example, the LCM equipment is showed in Figure 1, but in fact, it was not used in later experiment, which may be misleading. Additionally, it is better to marked the tissue thickness in your Figure 1 to give readers necessary information.

2. Some abbreviation errors exist in the paper, for example, the abbreviation of n-dodecyl- β -D-maltopyranoside is DDM but not DMM (Figure 1, L64 and L433).

3. Voxels number 2, 5, 8 were later suspended from the experiment due to suspicion to polymeric contamination, but the authors do not give explanation about what causes the contamination. Does the contamination introduce in the In-insert proteomics sample process procedure? If so, how can you avoid it?

4. L343 states the breast tissue voxels is 15 μ m thick, but L411 gives readers the information that the breast tissue was processed to a 14 μ m thick slices, what's the real thickness of the breast tissue used in the experiment?

5. In Figure 4, the authors investigate the protein post-translational modifications through the mass shift. We recommend them to provide corresponding MS/MS spectrum as example.

6. Citations were missing in some sentences. Like in line 245 – 246, "Furthermore, our results are consistent with 245 established blood biochemistry and proteomics".

Dear Reviewers,

Thank you for providing valuable feedback on our manuscript. We appreciate the thorough comments and we have carefully revised the manuscript to address your concerns. We have successfully addressed key points and would like to kindly request reconsideration of our revised version. Additionally, we kindly invite you to review our detailed responses to your suggestions. We started by addressing the questions from Reviewer 2 then Reviewer 3, and lastly Reviewer 1.

Addressing R2:Q1

I recommend rephrasing the justification for this study in the introduction. I agree that a method that does not rely on expensive equipment or reagents is advantageous. However, there are multiple statements that suggest this new method is better than these other approaches, which isn't the case (given the relatively small number of protein group identifications for the amount of tissue analyzed). I suggest that it would be better to focus the justification for this new method on the need for a simplified sample preparation approach that is comparable to current approaches, as this is still an advance for the field. I suggest rephrasing paragraph 2 of the introduction to account for this as how this paragraph is currently presented raises many questions such as: why is this method better for spatial proteomics than mass spectrometry imaging? Why is it better to use macrodissection rather than more precise microdissection using LCM? Can this method detect more protein groups than the others they are comparing to using the same amount of tissue? It becomes apparent further in manuscript that the proposed method is not better than previous ones, but instead has its own unique advantages while maintaining somewhat comparable protein group identifications, specifically with regards to accessibility and cost effectiveness (but not necessarily the proteomic data it can generate).

We agree with reviewer's assessment that stating our method is overall better than other approaches is inadequate. Therefore, we have removed any sentences in the text that would imply such superiority. PP 1, PP 2, PP 20-22. Instead, we now focus on comparability of our method to the current limited FFPE proteomics approaches. PP 2, PP 4-6

To provide stronger evidence of our methods potential as a viable alternative or complementary approach to current spatial FFPE methods, we have included and additional analysis implementing laser capture microdissection (LCM). We performed In-insert proteomics on 50 100 and 200 μm LCM dissected voxels of 5 μm thickness. This new data support our claim that our method offers the potential for improved spatial resolution beyond 1 mm^2 , especially when LCM microdissection is utilized instead of macrodissection. PP 4-5, Supplementary file 2, PP 33

We emphasized that In-insert method is suitable for low-budget experiments conducted in less-equipped labs. By introducing LCM combination with In-insert method, we have shown the versatility of our method, which is compatible with both LCM and macrodissection. Consequently, we did comment on the possible advantages of macrodissection over LCM. PP 4-5, PP 20, Supplementary file 2, PP 33.

In response to your suggestion we have added a comparison with a method utilizing the entire slide breast tissue slide, as published by Weke et al. [1] and with the publicly available TMT breast cancer

dataset [2] to further enhance the projection of our methods performance. PP 5, Supplementary file 3, PP 33

Considering your feedback that In-insert is complementary rather than superior to current methods, we have revised the manuscript to emphasize its own unique advantages. The main strengths lie in its cost-effectiveness and the reduced dependency on laboratory equipment. PP2, PP 6, PP 20

Addressing R2: Q2

2. The “spatial resolution” is quite low using this approach (based on regions that are between 2-10mm²). Given this, I would temper the references to the advantages of this approach for “spatial proteomics” as each of these regions would contain thousands of cells and other spatial proteomics approaches have a much higher resolution (e.g. mass spec imaging). I suggest that the authors acknowledge this as a limitation, but highlight instances where such an approach would be useful (e.g. tumor vs surrounding tissue perhaps).

To address this question/concern, we have made the following revisions:

We have included a determination of the In-insert method’s limit of detection or effectivity, as shown in Supplementary file 2 C and D. We have demonstrated that when the In-insert method is combined with LCM, it can achieve a spatial resolution of 0.0025 mm² (50 μm x 50 μm) in 5 μm thick slide while still identifying up to 450 proteins (FDR < 1%) per voxel. This performance is comparable to the autoPOTS method [3]. PP 4-5, PP 20. To further address your question, we added a note comparing IMS to In-insert proteomics in the discussion section PP 4-5, PP 20.

In addition, as per the reviewer's suggestion, we have emphasized the possible advantage of using macrodissection in conjunction with the In-insert method. By incorporating this aspect, we demonstrate the flexibility of our approach, which can accommodate both macrodissection and LCM techniques. PP 20

Addressing R2: Q3

3. The area of tissue that they are using to demonstrate their approach is quite large (2-10mm² regions from 15um thick sections) in comparison to other similar approaches using FFPE tissue (often regions half this size and sections <10um). Given this, it is difficult to determine direct evidence that their method generating better (or even comparable) results given that there are multiple other approaches in the literature capable of identifying similar/more protein groups using smaller amounts of FFPE tissue. Please remove/rephrase the statements suggesting their method is better than others that use a much smaller amount of starting tissue as these are not an equivalent comparison and do not support the superior nature of their approach (e.g. Intro: “Nevertheless, implementation of these protocols on 1mm² FFPE tissue led to identification of only several hundred proteins” and Discussion: “It has been shown that FFPE processing implementing MS-compatible detergents such as Rapigest, PPS, ProteaseMAX could identify up to 270 proteins out of 1 mm² 4 μm kidney FFPE slide”). Furthermore, I don’t agree that serial dilution after sample preparation is an equivalent comparison to previous studies that have analysed smaller amounts of tissue – one of the main roadblocks with these types of studies is sample loss when processing

microscopic amounts of tissue – diluting a larger amount of starting tissue post-processing does not address this issue and is therefore cannot be directly compared to other approaches that use a smaller amount of starting tissue. A better comparison would be analysis of the same size regions of tissue using their approach. If the authors wish to show evidence that their method works using a smaller area of tissue, then this must be done using a smaller area of starting tissue.

We have provided a direct evidence of the In-insert method's capabilities showing its ability to identify up to 450 proteins (FDR<1%) in a 5 µm thick voxel with of 0.0025 mm² (50 µm x 50 µm) area. In addition, we have included 100 and 200 µm lateral size voxels (Supplementary file 2 C and D). We believe that the including these new results will effectively address the reviewer's concern regarding the size of the smallest possible voxel area for the spatial In-insert FFPE proteomic analysis. Supplementary file 2, PP 4-5, PP 20.

As suggested, we have added a benchmarking analysis to other methods used for spatial FFPE proteomics showing complementary and comparable nature of the In-insert method to robotically assisted microPOTS in terms of the number of proteins identified from similar voxel sizes and thicknesses. We also benchmarked the In-insert method against nanoPOTS and TMT assisted spatial FFPE proteomics method demonstrating their superior performance over In-insert approach. PP 4-5, PP 20, Supplementary file 2, PP 33.

We agree with reviewer that serial dilution approach does not accurately represent the effectivity of the In-insert method. Therefore, we have included direct evidence from LCM dissected, In-insert processed voxels of FFPE normal breast tissue, with a thickness of 5 µm, as presented in the supplementary file 2 C and D instead of dilution series. As stated above, this new data now allow us to conclude that the In-insert method is more effective if compared to a study implementing mass spectrometry compatible detergents identifying 270 proteins out of 1 mm² 4 µm kidney FFPE slide. PP 20, Supplementary file 2 C and D, PP 33

Addressing R2: Q4 and R3Q3

4. I suggest including a comparison of their results to the known proteome of breast tissue (which could either be obtained from direct analysis of the same frozen tissue or more broadly from a comparative dataset in the literature that provides a more comprehensive analysis of breast tissue). This would be useful to see whether the proteins that are detected in this study are the most abundant ones in breast tissue or if there is an overabundance of common contaminants (as is often the case when analyzing small regions of FFPE tissue). This is important as many of the highlighted proteins in this study are keratins, which are also known contaminants – it would be useful to show that this is expected in breast tissue.

We appreciate the reviewer's concern regarding origin of keratins and potential contamination. To address this question, we have performed a comparison of the keratin spectral count ranks obtained using the In-insert method (N= 17 voxels) with those from normal breast FFPE tissue slides (N = 11 slides) processed by the method published by Weke et al.[1] and a publicly available dataset from TMT breast cancer FFPE proteomics (Supplementary file 3) The results of our analysis reveal that the keratins exhibit

similar intensity patterns across compared datasets. This finding strongly suggests that the presence of keratins is not due to the contamination but rather represent genuine biological signals. The data are illustrated as Supplementary file 3 and relevant comments on these findings were incorporated in the revised manuscript. PP 4-5, PP 21-22, Supplementary file 3, PP 33

Addressing R2: Q5

5. I recommend a clearer explanation of the advantage of in-insert analysis; this is currently unclear and is presented as a defining advantage of this protocol.

We have prepared a benchmarking experiment comparing the In-insert sample preparation method with autoPOTS, nanoPOTS and TMT FFPE spatial proteomics method. The result of this analysis have been included throughout revised manuscript. Our findings demonstrate that the In-insert method can serve as a viable cheap alternative to the robotically assisted autoPOTS platform. We have shown that the effectivity achieved with the In-insert method is comparable to that of autoPOTS. This implies that researchers who will prefer not to use robotically assisted platform could still achieve similar extraction effectivity in terms of proteins identified with In-insert method. Additionally, a new chapter “Defining the required amount of FFPE tissue and comparison to the current spatial FFPE tissue processing methods” has been added to the revised manuscript. This chapter provides a comparison of the In-insert method with other protocols currently available for spatial LC-MS/MS analysis of FFPE tissue. Through the revised manuscript we emphasize the complementary nature of the In-insert method in relation to existing spatial proteomics methods. PP 4-5, PP 20, Supplementary file 2 C and D, PP 33

Addressing R2: Q6

6. No complete dataset of results is provided in supplementary tables – only the top 11 most abundant proteins is included in Sup3. Please include a supplementary table of results showing to relative abundance of all protein groups between voxels. We acknowledge reviewer’s suggestion regarding including the full quantitation data from 17 voxels in our study. In our revised manuscript we stated that the complete quantitation data are available online in ProteomeExchange repositories. Considering the extensive nature of the full quantitation data, we have decided not to include a tabular form as a supplementary file. However, by providing the information on the availability of the full quantitation data in the ProteomeExchange repositories, we ensure that interested readers can access the data. We appreciate reviewers understanding of this decision and we believe that availability of the complete quantitation data in the ProteomeExchange is sufficient. PXD037609 under username: reviewer pxd037609@ebi.ac.uk and password: fq17pAeG as stated in paper on PP 27-28.

Addressing R3: Q1

1. The workflow in Figure 1 showed us the authors aim to develop a non-robotic In-insert FFPE proteomics approach to allow spatially resolved quantitative proteomics analysis of FFPE tissue in a user-friendly and economical manner. However, it seems to not a user-friendly or effective method, for it includes cumbersome procedure involving many manual processing steps, which is not user-friendly and may exist potential sample loss. In L30 the authors state that in other sample preparation methods “the detergents and chaotropes must be removed from the sample prior digestion or LC-MS analysis which often leads to excessive sample losses”, but more obvious sample loss may exist when the manipulator scraping each voxel and transfer tissue and lysis buffer into glass insert by using the In-insert FFPE proteomics approach.

We have removed the statements claiming that In-insert is user-friendly. Instead, we now emphasize its potential advantages in terms of cost effectiveness and minimal requirements for lab equipment. By focusing on these aspects, we highlight the practical benefits of the In-insert method compared to other spatial proteomics methods. PP 5-6, PP 20

We acknowledge reviewers concern about the potential sample loss during transfer to the insert. To address this, we have introduced a combination of the In-insert method with Laser Capture Microdissection (LCM). This combination allows for transfer of tissue voxels smaller than 100 μm lateral size while still obtaining sufficient MS signal. Our claims are supported by documentation in Figure 1 and Supplementary figure 1 D and E, Supplementary file 2 C and D, PP 33, demonstrating that the potential sample loss is minimized by reducing the number of processing steps and contact with the plastic surfaces. We have also added a section in the Materials and Methods describing the transfer of tissue voxels from LCM tube adhesive cap. Supplementary file 2 C and D could be considered as a proof that transfer of stakes or voxels could be done effectively as we could acquire sufficient TIC giving rise to almost 450 protein groups from a 50 μm lateral dimensions voxel. Additionally, we have mentioned that for more controlled handling with the sample, macrodissected stakes could be scraped and transferred under supervision using a microscope or magnifying glass, particularly during the transfer of LCM dissected voxels from adhesive cap. PP 4-5, PP 20, PP 23 Supplementary figure 1 D and E , Supplementary figure 2 C and D, PP 33,

Addressing R3: Q2

L44 the authors indicate the In-insert FFPE proteomics approach is a “novel, effective, user-friendly and economical ” method, but the sample preparation procedure seems to be a simple combination of other publications discussed in this paper, from which we could not see the novelty of this method. In Figure 2, the authors benchmark their approach, an important analysis that is missing here is about the quantitative reproducibility of their approach. Besides, the authors also need to provide the information about the area of each voxel and the corresponding protein identification depth, and how does this compare to other sample preparation protocols ?

We appreciate the reviewer's feedback regarding the perception that our protocol is a simple combination of previously published methods. We have taken this feedback into consideration and we have emphasized/added novel aspects of the In-insert sample preparation method in the revised manuscript version and FIG1.

One of the key aspects is the custom-made "wet chamber", which is depicted in FIG1. This wet chamber, to our knowledge, has not been reported before and enables relatively long (1 hour) incubation of FFPE voxel in lysis buffer at ~95°C which contributes to distinctiveness of our protocol. Furthermore, we emphasized the novelty of enabling the snap freezing, homogenization, sonification, centrifugation, high temperature de-crosslinking and also sample injection into the LC-MS system, all in single container-glass insert. This integrated workflow, combining multiple steps into a streamlined process, has not been previously presented in the context of FFPE proteomics. By emphasizing these unique aspects, we aim to clarify that the In-insert sample preparation is not merely a combination of existing techniques but harbors novel features. Additionally, we have highlighted the novelty of combining the In-insert sample preparation with label-free data independent acquisition (DIA) to generate protein spatial regulation heatmaps. This integration of the sample preparation and DIA further contributes to the originality and innovation of the In-insert method. To ensure clearer presentation of the novelty, we have incorporated statements emphasizing abovementioned unique aspects in both Introduction and Discussion sections, and additionally in FIG1. PP 2, PP 4-5, PP 20. We believe that the revisions now better express the novel aspects of the In-insert preparation method.

We have added a Supplementary file 5 showing the correlograms, protein heatmap and a PCA addressing the question on quantitative reproducibility of the approach. The relevant details can be found in PP 13 as well as in Supplementary file 5, PP 34 showing the overall good quantitative performance achieved with the In-insert - DIA quantitative mass spectrometry applied over processed FFPE tissue voxels.

We have addressed the reviewer's comment on the missing information regarding the area of macrodissected voxels in the manuscript. In response, we have made modifications to Supplementary file 1, now including the areas of voxels. We have also improved clarity of numbering of the voxels in the same figure. This allows easy matching of the voxels and their areas to the numbers of identified proteins presented in FIG2. To further compare the In-insert method with other spatial FFPE proteomics methods, we have conducted a benchmarking analysis as shown in previous responses. This analysis demonstrates the complementary and comparable nature of In-insert proteomics method to the robotically assisted autoPOTS methods in terms of the proteins identified from similar voxel sizes and thicknesses. The comparison to nanoPOTS and TMT-assisted spatial FFPE proteomics method, clearly showed their superior performance over the In-insert approach. We have summarized these findings in a new chapter titled "Defining the required amount of FFPE tissue and comparison to the current spatial FFPE tissue processing methods." The relevant details can be found in PP 4-6, PP 20 as well as in Supplementary files 1, 2, 3, PP 33. These additions and modifications provide a comprehensive comparison of the In-insert method with other existing spatial FFPE proteomics methods, highlighting its strengths and limitations as required by reviewer PP 4-5, PP 20.

Addressing R3: Q3

3. L95-101 and Supplementary file 2C the authors state “LC-MS/MS analyses of 30x, 90x, 300x and 1800x diluted samples suggest that the method also works for less than 1 mm² and thinner FFPE voxels”. Simply dilute the sample may not support the statement that the method can work for less than 1 mm² and thinner FFPE voxels. Directly process sample that less than 1 mm² may better support this statement and it is better for the authors to give us the information about the minimum area which can be processed by the In-insert FFPE proteomics approach.

We have incorporated additional results retrieved from LCM processed 5 µm thick voxels with lateral sizes of 50 µm, 100 µm, 200 µm. These results confirm the statement that our method is effective for smaller tissue voxels, as suggested by reviewer. PP 4-6, Supplementary file 2 C and D, PP 33

Addressing R3: Q4

4. In Figure 2C, the authors do not give enough explanation about why the DDA data identified more protein groups than DIA regardless of the search engine used, for DIA will offer superior depth in most cases. The detail information about the spectrum library built in this study for DIA is also missed in this paper.

Until relatively recently, searching DIA data was uncommon due to missing information about precursor m/z, making it less suitable for protein identification compared to DDA data. DIA data were primarily designed for quantitation purposes. However, an option to search DIA data has emerged and we included it in our study. This option is crucial especially for small voxels where only single injection to the LC-MS/MS system is available. In such samples both quantitation and spectral library building must be performed in single DIA run. Addressing the reviewers question, it is important to note that that less quantitative DDA methods rely on the intensity-based selection of top N precursors, making them less suited for accurate quantitation. However, these methods perform best in qualitative analyses and thus yield more protein identifications compared to DIA. While, it is more preferable and accurate to utilize DIA data for quantitative purposes rather than DDA data. DIA protein quantitation represents sophisticated method acquiring MS/MS spectra from all peptides entering the mass spectrometer. However, it might not be able to identify as many proteins as DDA. We have added a short note to clarify the reason for DDA identifying more proteins than DIA in Results section PP 7

We acknowledge that the information on building the spectral library is too brief. Therefore we rewired and clarified the details on building up the spectral library in the Materials and Methods section. PP 26-27

Addressing R3: Q5

5. In Figure 2, the authors compare DDA and DIA use different engines and draw the conclusion that MaxQuant significantly outperforms the Comet and MSFragger (L117). The authors also find DDA data in

MaxQuant with a MBR function is the most effective option for In-insert proteomics sample preparation compared with other searches (L107-109). But why the authors finally choose DIA combine with Skyline for further analysis which may be confusing for readers.

We acknowledge reviewers observation that readers outside the field of quantitative mass spectrometry might find it confusing why DIA data were used for protein quantitation across the voxels. Therefore we added a short statement at PP 7 clarifying why we choose to use DIA data for protein quantitation and DDA data for building spectral library. Please refer also to response to R3: Q4 if more details needed.

MINOR R3:

1. The workflow introduced in Figure 1 could not well depict your experimental procedure. For example, the LCM equipment is showed in Figure 1, but in fact, it was not used in later experiment, which may be misleading. Additionally, it is better to marked the tissue thickness in your Figure 1 to give readers necessary information.

We acknowledge reviewers observation regrading showing LCM equipment in FIG1 while not mentioning it in original manuscript. In revised manuscript we have incorporated LCM into our experimental approach, and as a result, we kept the scheme of the LCM equipment in the FIG1. Additionally, we have also included the FFPE tissue thicknesses into the FIG1. PP 4

2. Some abbreviation errors exist in the paper, for example, the abbreviation of n-dodecyl- β -D-maltopyranoside is DDM but not DMM (Figure 1, L64 and L433).

All DMM abbreviations unified and changed to DDM as suggested by reviewer (FIG1, PP 3, PP 23-24)

3. Voxels number 2, 5, 8 were later suspended from the experiment due to suspicion to polymeric contamination, but the authors do not give explanation about what causes the contamination. Does the contamination introduce in the In-insert proteomics sample process procedure? If so, how can you avoid it?

We have clarified how the contamination was introduced and gave 2 solutions how to avoid it. PP 22

4. L343 states the breast tissue voxels is 15 μ m thick, but L411 gives readers the information that the breast tissue was processed to a 14 μ m thick slices, what's the real thickness of the breast tissue used in the experiment?

We have unified the statements about the voxel thicknesses in the revised manuscript PP 20, 22. Voxel size was 15 μ m.

5. In Figure 4, the authors investigate the protein post-translational modifications through the mass shift. We recommend them to provide corresponding MS/MS spectrum as example. **We have included an annotated spectrum and fragmentation evidence for a lysine-methylated miscleaved peptide. The fragmentation spectrum of the NH₂-IAVAQYSDDVK_{methylated} (+14.0156 Da)VESR-COOH peptide demonstrates the detection of y and b fragment ion series on Exploris480, providing substantial evidence for the methylation of lysine at position eleven within the y product ion series (Supplementary file 4). We hope that including this spectrum fulfills reviewer's request. Supplementary file 4, PP 11, PP 33**

6. Citations were missing in some sentences. Like in line 245 – 246, “Furthermore, our results are consistent with 245 established blood biochemistry and proteomics”. **We have added a citations especially the missing one in line 245 – 246, “Furthermore, our results are consistent with 245 established blood biochemistry and proteomics”. As reviewer suggested. PP 14.**

Addressing R1:Q1

1. My main concern is that the authors did not compare this novel “In-insert FFPE” proteomics approach with a basic proteomic approach. Although a reference has been reported for an implemented mass spectrometry method for protein recovery from FFPE samples, such reference describes a method applied on kidney and not on breast tissue. For instance, I would have like to see a comparison of the results obtained from “In-insert FFPE proteomics” with those obtained from a basic proteomic approach on a consecutive slide of the same mammalian breast FFPE sample.

We addressed reviewer’s question and included comparison to concurrent spatial FFPE approaches in a new chapter titled “Defining the required amount of FFPE tissue and comparison to the current spatial FFPE tissue processing methods“. We added a benchmarking experiment to compare the In-insert protocol with other selected FFPE processing protocols. Initially, we compared the results obtained from macrodissected voxels processed via In-insert method to the FFPE proteomics of FFPE breast tissue slides (N=11) processed by the method by published by Weke et al. [1]. This method utilized acetonitrile and dithiothreitol in the extraction buffer and successfully identified approximately 1100 protein groups (FDR<1%) from each slide. This finding reveals the extractability of proteins from breast tissue and provides an estimate of the expected protein yield. Interestingly, our comparison shows that the In-insert method achieved a similar number of protein identifications from each voxel, demonstrating the effectiveness of our approach with an input size at least 20 times smaller if In-insert with macrodissection was utilized. However, after combining LCM with In-insert method and determining 50 µm lateral size as the smallest voxel compatible with In-insert, the benchmarking to the classical protocols becomes less meaningful since it becomes evident that In-insert is be more effective. Therefore, we have prepared more detailed benchmarking to concurrent microPOTS and nanoPOTS spatial FFPE proteomics. Additionally, we compared In-insert proteomics to FFPE spatial proteomics utilizing TMT tags. Furthermore, a comparison of keratins abundance in 3 datasets (In-insert processed breast FFPE voxels, preparation of entire breast FFPE slides by Weke et al., and TMT FFPE breast cancer dataset) reveals further insights into the performance of In-insert method compared to other methods. PP 4-6, PP 20, Supplementary file 2, 3, 5, PP 33-34.

We have provided further evidence that the In-insert method is a reliable and practical choice if researchers are seeking alternatives to robotically assisted FFPE proteomic platforms, particularly in low budget or less equipped laboratories. Moreover, we think that our findings determine the In-insert method for facilitating quick an unplanned spatial proteomic screening on FFPE slides. Importantly, we have shown that In-insert method delivers comparable effectiveness to microPOTS, offering a cost-effective solution that yields complementary protein identification results at a lower price. With this evidence we aim to persuade the reviewer that the In-insert method is a credible and advantageous approach for spatial proteomics. PP 4-6, PP 20, Supplementary file 2, 3, PP 33

MINOR R1

2. “Introduction” section:

In the sentence “This problem has partly been overcome by implementing mass spectrometry

compatible detergents such as Rapigest, PPS silent surfactant, ProteaseMax or by using direct trypsinization”, please include some references related at this matter. Here are some examples: PMID: PMC7900714, 26690073, 21280217

We have addressed the reviewer's comments regarding the references missing in the introduction. Now, on the PP 2 a references related to FFPE processing with the aid of the Rapigest, PPS silent surfactant, ProteaseMax or by using direct trypsinization were added, as reviewer suggested.

3. “Introduction” section:

“Implementing TMT labeling of FFPE subsections, followed by pooling and adding a booster channel substantially improves sensitivity of protein detection”, please define “TMT”

We defined TMT on page PP 2.

4. “Results” section:“FIG1 demonstrates the versatility of the glass insert, which allows repeated snap-freezing in liquid nitrogen, bath sonication, centrifugation and de-crosslinking at nearly 100°C followed by protease digestion at microliter scale.” None of the steps “repeated snap-freezing in liquid nitrogen, bath sonication, centrifugation” are indicated in Figure 1., please show or include those steps in the workflow of Figure 1.

We have addressed the reviewer's comment regarding the missing steps in FIG1. The steps in FIG1 now include snap-freezing in liquid nitrogen, bath sonication, centrifugation, de-crosslinking at nearly 100°C, and protease digestion at microliter scale. We have inserted the updated FIG1 in the revised manuscript text.

5. “Materials and methods”, “Study design” subsection: In the sentence “Voxels number 2, 5, 8 (Supplementary file 1C) were later suspended from the experiment due to suspicion to polymeric contamination.” It is very hard to see the number of the voxels in the supplementary file 1 B, C. This reviewer suggests numbering every voxel digitally.

We have addressed the reviewer’s comment on the missing digital numbering of the voxels in Supplementary file 1 A-C. In response, we have made modifications to Supplementary file 1, we improved clarity of numbering of the voxels in the figure A-C. This allows easy matching of the voxels and their areas to the numbers of identified proteins and other graphs presented in FIG2.

6. Supplementary Figure 2D: Why did the authors choose the 30x and not the 10X diluted sample to test the effect of gradient length on protein identification? In the 10x diluted sample a higher number of peptides was identified as showed in Figure 2C.

We agree with the reviewer's suggestion that using a 10X diluted sample would be better for testing the gradient length instead of 30X. However, we encountered a limitation in sample availability as we had already used it for multiple method optimization steps. Consequently, we only had access to a remaining 30X diluted sample for testing the gradient length. Nonetheless, we believe that the results still demonstrate the impact of decreasing gradient length on protein identification, even with the 30X diluted sample.

7. Figure 3 C, D, In the caption it is stated that the semi-tryptic peptides and the miscleaved peptides are considered across 17 voxels, but the figure shows 20 voxels, please clarify.

In this particular matter, we respectfully disagree with the reviewer. We counted the voxels presented in Figure 3C and D, and the count confirms the presence of 17 voxels. We mentioned in the text that three voxels were lost during the sample preparation process. The cause of this loss is explained on PP 22, along with suggestions on how to prevent such losses, as outlined on the same PP 22.

8. Do the authors know why only voxel number 6 and 19 showed a preference for tryptic cleavage at lysine? Did these voxels follow the same exactly sample preparation procedure of the other voxels?

We appreciate the reviewer's observation, but we do not have a definitive explanation for this finding. We can only speculate that the formalin crosslinking during sample fixation might not have uniformly affected the tissue, leading to differences in trypsin access to lysines in voxels 6 and 19 compared to other voxels. Additionally, we noticed a slight shift in the modification landscape in sample 19 compared to the other voxels. However, we consider it improbable that the discrepancy can be attributed to the In-insert sample preparation method, as all the samples were processed equally in a single batch

9. Figure 6. The authors stated that type I cytoskeletal 10 (K1C10) protein is closely associated to PIP, however K1C10 is neither represented in the network of Figure 6A nor as a spatial protein intensity heatmap like the other keratins. On the other hand, there is a discrepancy with the caption of Figure 6, where it is stated that panel E is the spatial heatmap of K1C10, but in the figure it is represented by the spatial heatmap of K1C18. Please correct.

We appreciate the reviewer's observation regarding the discrepancy in the figure legend, specifically related to K1C10 and K1C18. In order to align the logic of FIG6 legend with the manuscript text and FIG6, we have made the following modifications:

Firstly, we have decided to retain the K1C18 spatial heatmap in FIG6. We have verified that this heatmap accurately corresponds to the protein K1C18. Secondly, we have addressed the discrepancy by correcting the contents of the figure accordingly. PP 18, PP 19

Unfortunately, due to space limitations in Figure 6, we were unable to include K1C10. However, we would like to emphasize that the close spatial regulatory trends of K1C10 to the PIP observed across the slide are included in Supplementary File 9. To make it more apparent, we have highlighted the proteins displaying the same spatial regulation trends using the color red, as mentioned on PP 16.

We hope that these modifications have resolved the issue.

10. "Discussion" section: "PIP may prove particularly valuable for screening breast tumor regions..." This result comes from the analysis of only one sample of breast cancer, the authors should discuss that this result needs to be confirmed in a larger number of samples.

We acknowledge the reviewers comment and completely agree that relying only on In-insert proteomics data from 17 voxels in one slide alone does not provide sufficient evidence for PIP to be a biomarker for screening breast tumor regions. To partially address this issue we have included a result from KMplotter analysis. This analysis has revealed a potential association between PIP and breast cancer, as KMplot in Supplementary file 11 indicates that high expression of PIP at the protein level in breast cancer patients (N=126) is significantly correlated with lower overall survival ($p=0.054$) within a

120 month post diagnosis period. These results emphasize importance of further research to fully elucidate the role of PIP in breast tissue. PP 21-22, Supplementary file 11, PP 35

In addition we added to discussion a statement that further validation must be performed to draw any conclusions on if PIP may prove particularly valuable for screening breast tumor regions.

Literature:

- [1] K. Weke *et al.*, "DIA-MS proteome analysis of formalin-fixed paraffin-embedded glioblastoma tissues," *Anal. Chim. Acta*, vol. 1204, p. 339695, Apr. 2022, doi: 10.1016/j.aca.2022.339695.
- [2] K. Asleh *et al.*, "Proteomic analysis of archival breast cancer clinical specimens identifies biological subtypes with distinct survival outcomes," *Nat. Commun.*, vol. 13, no. 1, Art. no. 1, Feb. 2022, doi: 10.1038/s41467-022-28524-0.
- [3] A. J. Nwosu *et al.*, "In-Depth Mass Spectrometry-Based Proteomics of Formalin-Fixed, Paraffin-Embedded Tissues with a Spatial Resolution of 50–200 μm ," *J. Proteome Res.*, vol. 21, no. 9, pp. 2237–2245, Sep. 2022, doi: 10.1021/acs.jproteome.2c00409.

Reviewers' comments:

Reviewer #1 (Remarks to the Author):

The authors did an excellent job responding to my questions. They addressed satisfactorily all concerns I had in my initial review. I find this manuscript acceptable for publication.

Reviewer #2 (Remarks to the Author):

I appreciate the considerable effort the authors have gone to rework this paper in response to reviewer feedback. Their current version nicely justifies the utility and need for this new proteomics approach and provides a large amount of new evidence about how their approach compares to other approaches. I have no further concerns or suggestions.

Reviewer #3 (Remarks to the Author):

The authors have addressed most of my comments and performed additional experiments using the LCM technology. However, the authors didn't properly present and discuss the data in the main manuscript. In addition, the authors should perform side-by-side comparison of their method and discussed method, for example, processing the same sample using two methods. Otherwise, it is hard to conclude how the all-in-one method will avoid the contamination of high-resolution MS, especially for these DDM and staining dyes.

Reviewer 3:

The authors have addressed most of my comments and performed additional experiments using the LCM technology. However, the authors didn't properly present and discuss the data in the main manuscript. In addition, the authors should perform side-by-side comparison of their method and discussed method, for example, processing the same sample using two methods. Otherwise, it is hard to conclude how the all-in-one method will avoid the contamination of high-resolution MS, especially for these DDM and staining dyes.

Response:

We thank to reviewer for feedback on our revised manuscript and we value very important additional concerns that were raised. Here, we would like to introduce our response to these concerns, which has been included in the current version of manuscript attached.

1. We agree with Reviewer's concern of not presenting the data in the result section properly. Therefore, in Results section we have included a new table (Table 1) for side-by-side comparison as suggested (Page 5-6, line 96-97). It now better compares the acquired data from the benchmarked protocol by Weke et al. ¹ with variants of our In-insert protocol namely a combination with macrodissection and LCM microdissection of both stained and unstained sample. We thank reviewer for raising this concern as now reader could better choose the strategy that will be used for In-insert processing, and without first performing the laboratory work researcher could estimate the yield/result by inspecting the result from current study on breast tissue. Further, for increasing the conciseness and the structure of result section, we have moved part discussing the benchmarking to the intersecting methods based on literature search to the discussion section (Page 21, line 423-459). We believe that this was also one of the reviewers concerns, as the information seemed unnecessarily duplicated in both results and discussion sections. We have made the text in the paragraph "Defining the required amount of FFPE tissue and benchmarking variants of the In-insert method to the intersecting FFPE tissue processing method" (Page 4, line 81-96) more concise and have moved the fact from the text to the presented Table 1 (Page 5-6, line 96-97). In results section, minor changes have been introduced also to the paragraph "Spatial label-free DIA quantitative proteomics of FFPE breast tissue voxels" (removed details about DIA method which rather belong to Materials and methods) (Page 14, line 272) to increase the conciseness and readability of the text. In addition, some unnecessary statements in results section have been removed due to exceeding permitted word count. We believe that reviewer will now find the results section more concise and attractive to general public.
2. We appreciate that reviewer spotted that our discussion was not properly constructed and missed the ease of readability and included several mistakes. We have focused on rebuilding the discussion in more concise way especially from the point of removing the redundancies and concatenating them into continuous more readable text (Page 21-23, line 394-514). These changes relate mostly to properly presenting the new LCM results and macrodissection results and benchmarking them to data retrieved by Weke et al. ¹ protocol and to data from literature search of other intersecting methods (Page 21-23, line 415-459). In addition, some unnecessary statements in discussion section have been removed due exceeding permitted word count. We hope that Reviewer will now find the results section more concise and attractive to general public.

3. We appreciate important Reviewer's objective raised on behalf of hematoxylin staining, as if it would not be used properly or with certain limitations it could lead to poor performance of protocol or no results at all. Therefore, we have included a new paragraph in the discussion which discusses the limitations to mass spectrometry imposed by hematoxylin staining. We have highlighted studies demonstrating that LCM voxelation (small voxels ~50 μm lateral size) of stained FFPE tissue does not significantly affect LC-MS/MS data, despite chemical contamination, matrix effects and increase in background noise associated with hematoxylin staining. However, we have stated that we do not recommend In-insert sample processing of hematoxylin stained FFPE tissue in combination with macrodissection as here the contamination could be vast leading to signal suppression. In addition, we warn the reader that the distribution of hematoxylin staining is unequal across a tissue slide, thus complications might occasionally emerge even with smaller hematoxylin stained voxels retrieved by LCM microdissection ((Page 22, line 460-473) and Table 1, (Page 5-6, line 96-97)).
4. We appreciate also Reviewer's comment on behalf of DDM detergent, as it might have been unclear to reader that there are certain limitations in compatibility of DDM with LC separation and MS/MS analysis. In the initial paragraph of the discussion we have added an information from literature search that reveal to the reader the limits on DDM concentration allowed in samples for direct injection to mass spectrometer and we benchmark it to the DDM concentrations used in In-insert processed samples (Page 21, line 402-408).

Point 3. and point 4. should collectively address the Reviewer's concern on behalf of potential contamination of high resolution MS instrument. Taken together, the presence of DDM detergent and hematoxylin FFPE tissue staining could potentially be major contributors to this issue, as reviewer stated. (Page 21, line 402-408), (Page 22, line 460-473) and Table 1, (Page 5-6, line 96-97).

5. To further boost proper presentation of the data we have reconsidered estimation of the cell count for mammary tissue acini dissected in 50 μm x 50 μm FFPE voxels and have added cell count estimations for macrodissected voxels and for tissue slides in Table 1, (Page 5-6, line 96-97). In abstract, we have stated that our In-insert method in combination with LCM could identify up to 450 protein groups (FDR<1%) in voxels including tens of the cells rather than hundreds as stated in original manuscript version (Page 1). We supplement this statement with updated figure Supplementary figure 3F (hematoxylin stained mammary acini) which is also mentioned in the discussion (Page 21, line 420-421). We believe that reviewer will find this correction vital for more proper data presentation in manuscript.

REVIEWERS' COMMENTS:

Reviewer #3 (Remarks to the Author):

The authors have addressed all my concerns.